# An Integrated Approach for Analyzing the Morphological Evolution of the Lower Reaches of the Minjiang River Based on Long-Term Remote Sensing Data

**Nie Zhou** [1], **Sheng Sheng** [1,*], **Li-Ying He** [1], **Bing-Ru Tian** [1], **Hua Chen** [1] and **Chong-Yu Xu** [2]

1   State Key Laboratory of Water Resources and Hydropower Engineering Science, Wuhan University, Wuhan 430072, China; niezhou@whu.edu.cn (N.Z.); chua@whu.edu.cn (H.C.)
2   Department of Geosciences, University of Oslo, N-0316 Oslo, Norway
*   Correspondence: shengsheng@whu.edu.cn

**Abstract:** Understanding the evolution of river morphology is crucial for comprehending changes in water resources and implementing development projects along rivers. This study proposes an integrated approach utilizing remote sensing image data combined with deep learning and visual interpretation algorithms to analyze continuous-type changes in river morphology. This research focuses on the lower reaches of the Minjiang River in China and comprehensively analyzes the river's morphological evolution from 1986 to 2021. The results show that the proposed method of river water identification in this study demonstrates high accuracy and effectiveness, with an F1 score and Kappa coefficient greater than 0.96 and 0.91, respectively. The morphology of the river channel remains stable in the upstream and estuarine sections of the study region while undergoing substantial alterations in the middle section. Additionally, this study also identifies several factors that significantly impact the evolution of river morphology, including reservoir construction, river sediment mining, river training measures, geological conditions, and large flood events. The findings of this study can provide some insights into the management and conservation of water resources.

**Keywords:** river morphology; water body identification; remote sensing; UNet; MobileNet; Minjiang River

## 1. Introduction

Water is a crucial natural resource that underpins human life and well-being [1,2]. To accomplish sustainable, high-quality developments of socio-economic and environmental systems, water resources management problems must be prioritized, their status must be systematically monitored, their allocation must be optimized, and they must promote sustainable growth [3,4]. Rivers, as the main carriers of surface water resources, undergo morphological changes due to regime shifts in water resources [5], which have impacts on ecological environments, hydraulic engineering, and hydrological meteorology [6]. Monitoring rivers and examining their morphological evolution can enable us to better understand water resource trends and develop more effective river-based engineering projects, thereby promoting efficient water use, river regulation, and societal stability [7].

Traditional field surveys offer high precision but require significant effort and resources to measure complex river morphology, hindering continuous, high-temporal-resolution monitoring. Remote sensing images, however, provide extensive coverage, frequent revisits, and low-cost measurements, enabling periodic observations of surface features [8–11]. In hydrological remote sensing, these images have been widely employed for monitoring coastal changes and large lakes [12,13].

The launch of satellites, such as Landsat, Sentinel, SkySat, and SuperView, has significantly improved the spatial resolution of remote sensing imagery [14]. Specifically, the SkySat-1 and SkySat-2 have a GSD of 0.86 m for panchromatic bands and 1.1 m for

multi-spectral bands, while the SuperView-1 captures images with a 0.5 m resolution for the panchromatic channel, and 2 m resolution for the blue, green, red, and NIR multi-spectral channels [15,16]. Unfortunately, these satellites' data are not currently available for free. On the other hand, the Landsat and Sentinel satellite series provide high-resolution imagery data with spatial resolutions of 30 m and 10 m, respectively, which offer detailed topographical, vegetative, and hydrological information [17]. Furthermore, these data are available to the public for free, making them more suitable for broad scientific research. Through digitizing and analyzing remote sensing imagery, researchers can identify and analyze the geomorphic features of rivers, allowing for effective monitoring and analysis of the process of a river's morphology evolution. However, the traditional methods used to identify water bodies have limitations, including low accuracy in single-band threshold methods, complex calculations involved in multi-spectral band methods, and difficulties in determining water body boundaries when they appear fuzzy using water index methods [18]. The rise of machine learning algorithms offers innovative approaches for water body recognition and river morphology analysis [19–22]. With modern computers' powerful processing capabilities, high-precision automatic interpretations of remote sensing images are achievable, and the introduction of residual connection mechanisms further propels image recognition into deep learning territory. Deep neural networks, utilizing large-scale parameters for data fitting and training, can surpass humans' visual interpretation ability with sufficient data [23].

The Minjiang River basin, located in Fujian Province, China, is a major inlet to the East China Sea [24]. Since the 1990s, significant downstream changes have occurred due to the Shuikou Reservoir completion and rapid socio-economic development [25–27]. These changes manifest as altered water and sediment conditions, hydrodynamic conditions, and impacts on flood control, water supply, and coastal socio-economic development.

For the first time, this study proposes and applies an approach combining remote sensing images (1986–2021), deep learning algorithms, and visual interpretation techniques to examine the morphological evolution and underlying causes of the Minjiang River's downstream region over the past 35 years. This study enables us to enhance our comprehension of the river's present condition and future trajectories, systematically track variations in water resources, and provide crucial data to support river management initiatives.

## 2. Study Area and Based Data

### 2.1. Study Area

The downstream area of the Minjiang River is located in Fuzhou City of Fujian Province, China (Figure 1), spanning between 25°15′ and 26°39′N in latitude and 118°08′ and 120°31′E in longitude. This region is characterized by a subtropical monsoon climate with warm temperatures and abundant rainfall, averaging between 1400 and 2300 mm annually [28]. The study area encompasses the main stream of the Minjiang River, which flows from the Shuikou Reservoir to the sea, passing through Minqing (MQ), Houguan (HG), Mawei (MW), and Tingjiang (TJ) counties. Fuzhou City covers an area of 11,575 km², and the region had an estimated population of 7.34 million in 2019. The downstream channel of the Minjiang River is characterized by a complex river network, abundant water resources, and diverse ecological environment, making it an important water resource in the region.

The schematic diagram and the DEM of the study area are shown in Figures 1 and 2, respectively. The MQMH section is the river section between Shuikou Reservoir and HG; the rivers are mostly mountainous type rivers, which are relatively less directly influenced by human activities. In the vicinity of HG, the Minjiang River is divided into two sections: the NG section and the BG section. The BG section passes through urban areas and serves as a typical waterway for urban locales. In contrast, the NG section has a relatively wider river channel with a slower river flow. Subsequently, the NG and BG sections converge near MW, causing the river flow to change direction from southeast to northeast. After passing through TJ, the river is separated by Langqi Island before flowing into the East China Sea via the Meihua waterway and Changmen waterway.

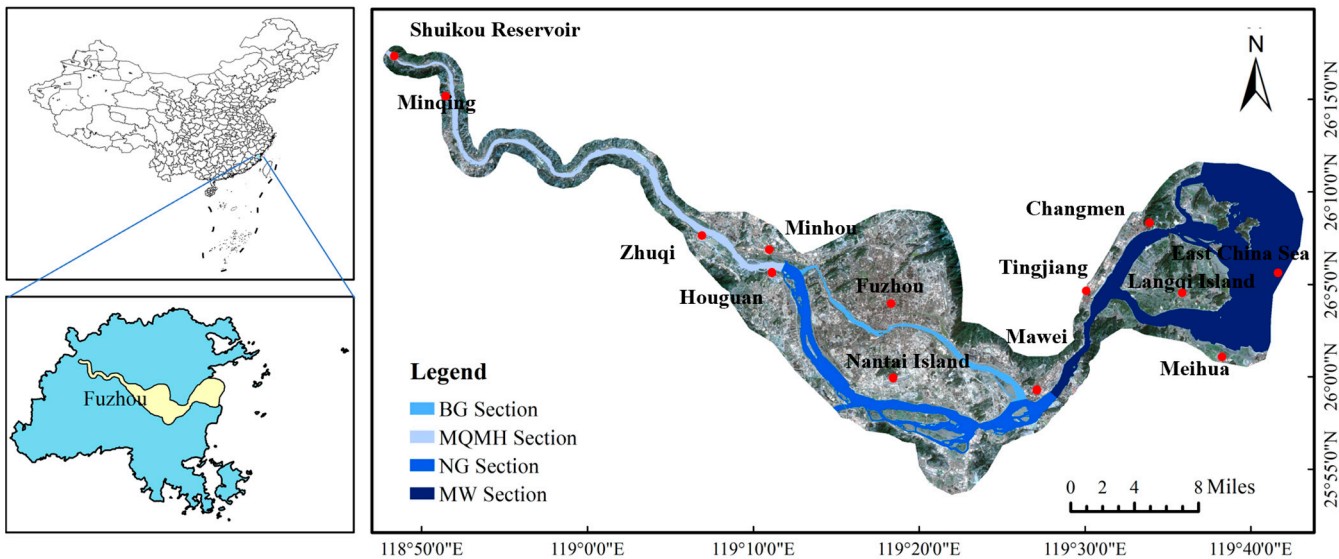

**Figure 1.** Schematic diagram of the study area.

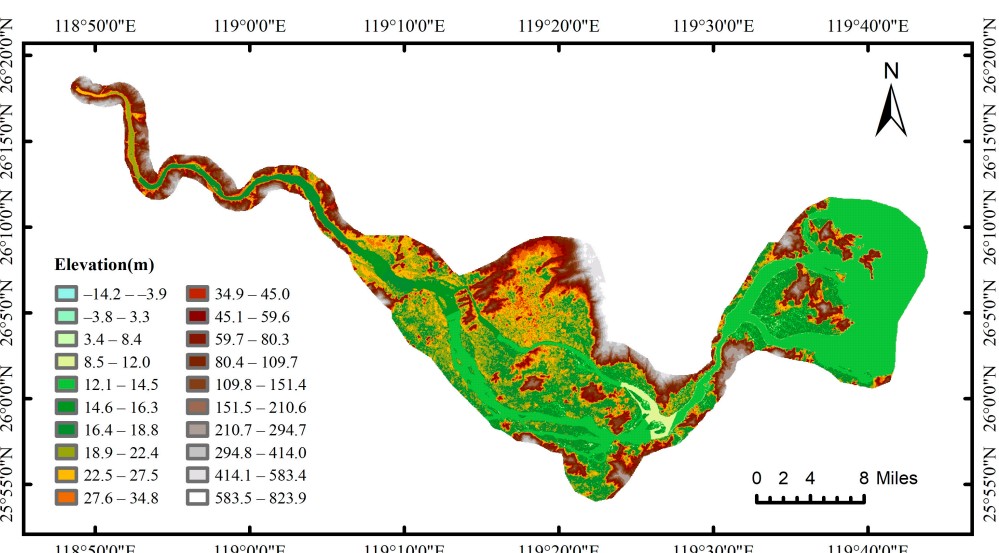

**Figure 2.** The DEM of the study area.

The vast upstream watershed and the influence of frontal rainfall make floods a frequent occurrence in this watershed. Since 1986, the watershed has experienced significant floods in 1992, 1998, 2002, 2005, 2006, and 2010. These floods resulted in a significant loss of lives and property, with the flood peak flow at Zhuqi hydrological station reaching 33,800 m$^3$/s on several occasions. In light of these challenges, comprehending the evolution of the river morphology of the Minjiang River is crucial for rational planning and development of water resources, as well as supporting the regulation of the river in the downstream section of the Minjiang River.

*2.2. Based Data*

2.2.1. Remote Sensing Image Data

In this study, remote sensing images from Landsat 5, Landsat 8, and Sentinel 2 satellites were used to study the evolution of river morphology in the lower Minjiang River, covering the period from 1986 to 2021. Due to the combined effects of satellite operating time and weather conditions, remote sensing images may be subject to interference, such as cloud cover and uneven lighting. Therefore, we screened the collected remote sensing image data to ensure their accuracy and reliability. Additionally, April data were specifically chosen

to ensure temporal consistency and avoid the impact of occasional factors, such as floods. Ultimately, 14 complete remote sensing image data were obtained (Table 1).

**Table 1.** Remote sensing image statistics table.

| Satellite Sensors | Time | Spatial Resolution (m) |
|---|---|---|
| Sentinel2 | 1 April 2021 | 10 |
| Sentinel2 | 7 April 2019 | 10 |
| Landsat8 | 18 April 2017 | 30 |
| Landsat8 | 13 April 2015 | 30 |
| Landsat8 | 23 April 2013 | 30 |
| Landsat5 | 2 April 2011 | 30 |
| Landsat5 | 12 April 2009 | 30 |
| Landsat5 | 4 April 2006 | 30 |
| Landsat5 | 12 April 2003 | 30 |
| Landsat5 | 3 April 2000 | 30 |
| Landsat5 | 11 April 1997 | 30 |
| Landsat5 | 19 April 1994 | 30 |
| Landsat5 | 21 April 1990 | 30 |
| Landsat5 | 29 April 1986 | 30 |

### 2.2.2. Model Training Dataset

To train our model, we sourced water body samples from two different methods: geographic country survey (China) and manual annotation. The geographic country survey data are overlaid onto remote sensing images of varying resolutions using ArcGIS software. After careful inspection and refinement, the resulting vector data are converted into a binary mask map. Similarly, manually labeled images are created using ArcGIS software to generate water body vector data, which are converted into binary mask maps and cropped into 256 × 256 size training data.

To improve the model's recognition ability under diverse conditions, we expanded the dataset using image enhancement techniques such as random rotation, contrast adjustment, horizontal and vertical flipping, and brightness contrast adjustment [29,30]. By combining these data pre-processing steps, we obtained a total of 138,000 and 61,380 sets of Landsat and Sentinel data, respectively. These were then divided into training and test sets at a 9:1 ratio.

## 3. Method

### 3.1. Remote Sensing Image Pre-Processing

The pre-processing of remote sensing images includes several steps. First, area network leveling is performed using the panchromatic image to match connection points and ensure an error of fewer than 3 pixels (Landsat-8 images only). Orthorectification is then carried out using surface control points and resampling the image into an orthophoto, utilizing Eurasian 30 m DEM data. The Landsat and Sentinel-2 sensor bands are then resampled to a resolution of 30 m and 10 m, respectively, using the nearest neighbor interpolation method. To achieve overall color balance, the hue, saturation, contrast, and brightness are adjusted based on grayscale characteristics of the remote sensing image and reference image. Color uniformity is achieved by applying a color uniformity template to each image block to avoid interpretation errors caused by excessive color differences during interpretation. Finally, through digital processing such as geometric mosaic, tonal adjustment, and de-overlap, a new mosaic network is generated, and mosaic splitting is performed to stitch the simultaneous remote sensing images into a complete image.

### 3.2. River Water Identification Method

Remote sensing image interpretation is carried out by combining neural network and visual interpretation method in this study. Firstly, the neural network utilizes multi-band data from remote sensing images as inputs and is trained with water and non-water labels

to automatically recognize water bodies. Specifically, the bands 1–5 and 7 of Landsat 5 images, bands 2–7 of Landsat 8 images, and bands 2–5, 9, and 10 of Sentinel-2 images are used as inputs, respectively. Additionally, based on high-definition remote sensing images, visual interpretation method is used for secondary correction of local areas with poor automatic recognition effects to improve the recognition effect of water bodies. Finally, GIS tools are used to calculate the morphological information of characterized rivers.

The water body identification in Landsat remote sensing images uses the UNet neural network. UNet is a highly symmetric encoder–decoder network widely used in image recognition fields such as biomedical, unmanned, and feature classification [31]. The model encoder layer comprises two convolutional layers of $3 \times 3$ size to extract features, and a $2 \times 2$ size maximum pooling module is employed to down-sample the extracted features. Each module in the decoder layer incorporates an up-sampling operation, which first employs a depth-space transformation through a $2 \times 2$ sized deconvolution module, and constructs a residual convolution layer connection at the same level to enable fusion of features across different scales. Finally, the decoded code is mapped to the water body identification result via a fully connected layer. Schematic diagram of UNet network structure as shown in Figure 3.

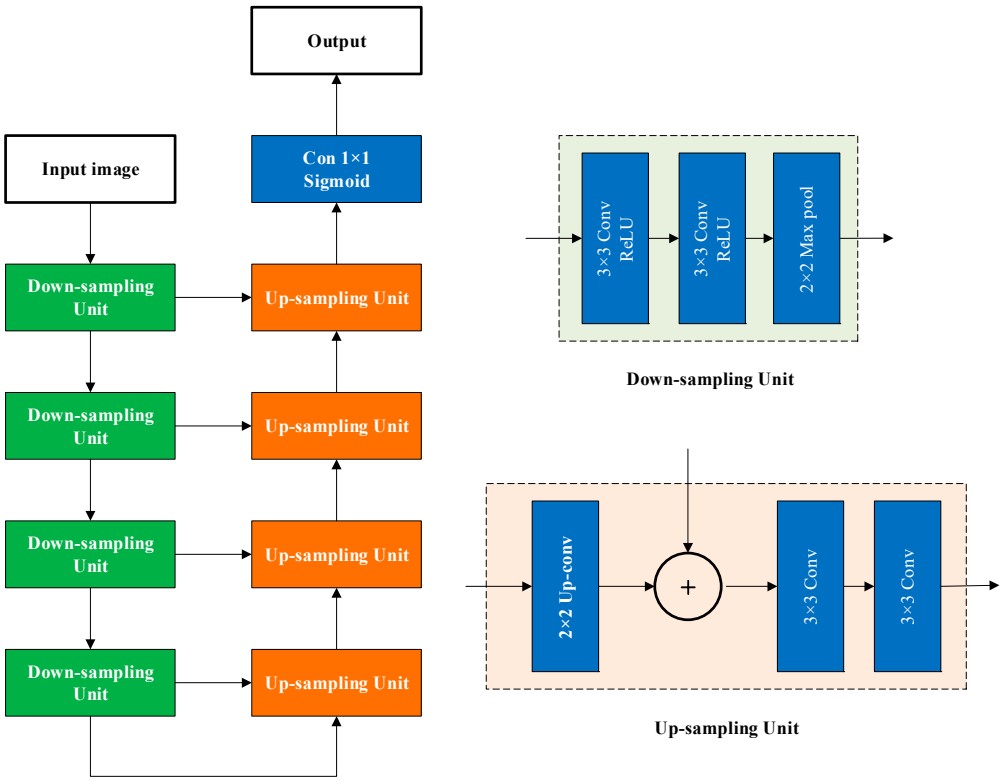

**Figure 3.** Schematic diagram of UNet network structure.

The identification of water bodies in Sentinel 2 remote sensing images utilizes a combined MobileNetV2 backbone network and DeeplabV3+ network proposed by Luo et al. [32] (MobileNetV2 is a lightweight and efficient convolutional neural network that achieves high accuracy with fewer parameters, making it suitable for real-time processing of remote sensing images). On the other hand, DeeplabV3+ is a powerful semantic segmentation model that accurately classifies and segments objects in images, thanks to its encoder–decoder architecture with deconvolution and skip connections that capture multi-scale contextual information and enhances spatial resolution. Combining these two networks results in a high-accuracy model with low computational costs. The MobileNetV2 backbone network efficiently extracts features, while the DeeplabV3+ network leverages the feature maps for accurate semantic segmentation. This combination is particularly useful for real-time

processing of large remote sensing datasets, specifically for water body identification. The specific structure of the proposed model is shown in Figure 4.

**Figure 4.** The combined model based on MobileNetV2 and DeepLabV3+.

### 3.3. Beaches and Sandbars Identification Method

Distinguishing sandbars and beaches from adjacent riverbanks and land in multi-band images can be challenging due to their various shapes. To address this issue, we utilize a visual interpretation method to manually delineate the range of sandbars and beaches in the study area, resulting in more precise edge curves. Landsat 5 and Landsat 8 images are then extracted using a combination of bands 543 (band 5 for red, band 4 for green, and band 3 for blue in RGB), while Sentinel remote sensing images are identified using bands 654 (band 6 for red, band 5 for green, and band 4 for blue in RGB). Beaches are typically found in curved portions of the river and appear as gray-white with high brightness. In contrast, sandbars of different sizes and shapes are distributed throughout the river channel and display a dark red color.

### 3.4. Evaluation of the Effect of Remote Sensing Image Interpretation

In order to comprehensively evaluate the effect of remote sensing interpretation, accuracy, precision, recall, F1 index, and Kappa [33] coefficient are used to analyze the effect of water body identification in this study.

Accuracy is the proportion of correctly classified samples to the total number of samples, and a value closer to 1 indicates better model performance. It is useful when the classes are well-balanced and the cost of false positives and false negatives is similar. Precision is the proportion of true positive samples to the total number of predicted positive samples, and it is a useful metric when the cost of false positives is high. Recall is the proportion of true positive samples to the total number of actual positive samples, and it measures the ability of the model to identify positive samples. It is useful when the cost of false negatives is high. F1 score is the harmonic mean of precision and recall, and it provides a balanced evaluation of the model's performance. It is particularly useful when the classes are imbalanced. Kappa coefficient is a statistic that measures the agreement between the predicted and actual classifications of a model, taking into account the possibility of agreement by chance. The value of kappa ranges between −1 and 1, with a value of 1 indicating perfect agreement and a value of 0 indicating agreement by chance.

Equations (1)–(5) can be used to express accuracy, precision, recall, F1 score, and Kappa coefficient, respectively.

$$Accuracy = \frac{TP + TN}{TP + TN + FP + FN} \tag{1}$$

$$Precision = \frac{TP}{TP + FP} \tag{2}$$

$$Recall = \frac{TP}{TP + FN} \tag{3}$$

$$F1 = \frac{2 \times Precision \times Recall}{Precision + Recall} \tag{4}$$

$$\begin{cases} p_0 = \frac{TP + TN}{TP + FP + FN + TN} \\ p_e = \frac{(TP + FN) \times (TP + FP) + (TN + FN) \times (TN + FP)}{(TP + FP + FN + TN)^2} \\ K = \frac{p_0 - p_e}{1 - p_e} \end{cases} \tag{5}$$

where $F1$ represents the $F1$ score; $K$ represents the Kappa coefficient; $TP$ represents the number of positive samples classified correctly; $TN$ represents the number of negative samples classified correctly; $FP$ represents the number of positive samples classified incorrectly; and $FN$ represents the number of negative samples classified incorrectly.

*3.5. River Morphology Change Parameters*

In this study, curvature coefficient, fractal dimension, land use transfer matrix, and the area change in sandbars and beaches are selected to characterize the morphological evolution of the river.

3.5.1. Curvature Coefficient

The curvature coefficient is used to characterize the degree of river curvature, with a higher value indicating greater curvature, which can be mathematically expressed using Equation (6).

$$K = \frac{L}{LR} \tag{6}$$

where $L$ is the length of the centerline of the river section; $LR$ is the straight-line distance from the beginning to the end of the river section; and K is the curvature coefficient of the river surface. Meanwhile, for the river with many branches and complex morphology, the curvature of the primary channel is mainly considered in this study, i.e., only the length of the primary channel is calculated, so as to characterize the curvature of the primary channel.

3.5.2. Fractal Dimension

The fractal dimension is used to describe the complexity and irregularity of the river morphology [34]. Box counting method was used to calculate the fractal dimension of the river. Different grid lengths of square fishing nets were created using ArcGIS software, and the nets were cut by the river surface file; after that, counting the number of nets obtained by cutting, the fractal dimension calculation can be expressed by Equation (7).

$$FD = -\lim_{r \to 0} \frac{\lg N(r)}{\lg(r)} \tag{7}$$

where $N(r)$ is the minimum number of grids covering the river in the corresponding year; $r$ is the size of the square grid covering the river; and FD is the fractal dimension of the river. By varying the size of $r$, $\lg N(r)$ is linearly related to $\lg(r)$, and FD is the absolute value of the slope of the line after linear regression of $\lg N(r)$ and $\lg(r)$.

### 3.5.3. Land Use Transfer Matrix

The land use transfer matrix is a valuable tool for analyzing changes in land cover over time in a specific area. It is often represented as a 2D matrix. By integrating the land use transition image with the transfer matrix, it is possible to identify the location and area of land type conversion between two different times. This approach allows for the observation of both the static distribution of various land types in a fixed area and the dynamic changes in the area of each land type over time. As a result, the land use transfer matrix is widely used in analyzing land use change [35,36].

This study first obtains the distribution of river water, riverbanks, and sandbars in each year through remote sensing image identification. The land use transfer image is then created using ArcGIS software, and the transfer matrix is constructed by calculating the conversion area between each land block type. The mutual conversion process between different land use types is subsequently quantitatively analyzed using the transfer matrix. The transfer matrix calculation is represented by Equation (8).

$$S_{ij} = \begin{bmatrix} S_{11} & S_{12} & \cdots & S_{1n} \\ S_{21} & S_{11} & \cdots & S_{21} \\ \vdots & \vdots & \ddots & \vdots \\ S_{n1} & S_{n2} & \cdots & S_{nn} \end{bmatrix} \tag{8}$$

where $S$ denotes the area of each land use type, while $i$ and $j$, respectively, represent the land use type before and after conversion. The total number of land use types is denoted by $n$.

## 4. Results

### 4.1. Reliability Assessment of Water Body Identification Algorithms

The study randomly deploys 1200 points to quantify the effects of remote sensing image spatial resolution on river water body identification. Among these points, 800 are identified as water bodies and 400 as non-water bodies through visual inspection, as displayed in Figure 5.

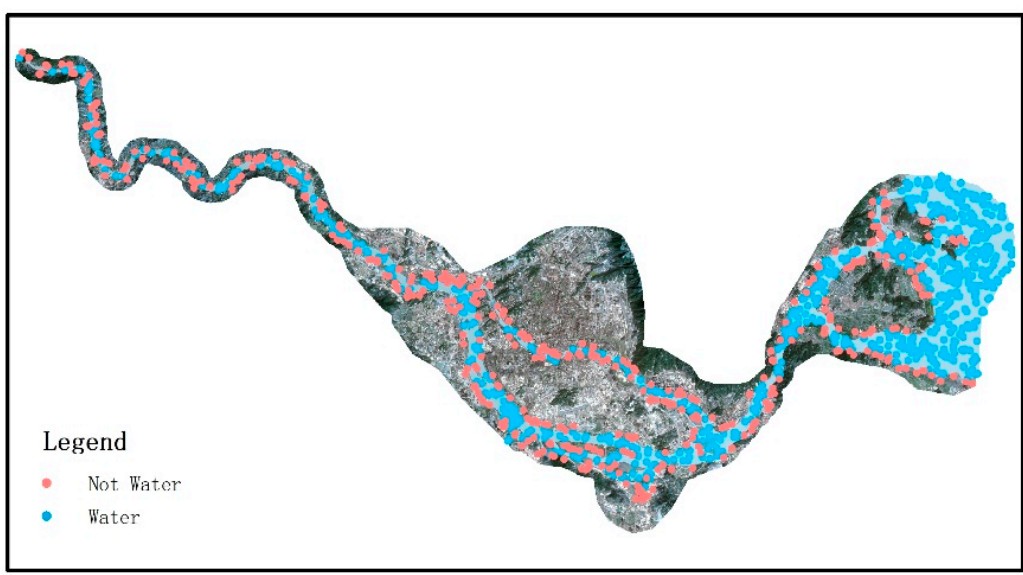

**Figure 5.** Schematic diagram of random sites in the study area.

Table 2 displays the accuracy, precision, recall, $F1$ index, and Kappa coefficient of the identification results of the remote sensing images. The $F1$ score of Sentinel-2 image exceeds 0.97, and the recall rate surpasses 0.95. The accuracy, recall, and $F1$ index of Landsat-8 image are 0.960, 0.943, and 0.969, respectively. Furthermore, the Kappa coefficients of

Sentinel image and Landsat image are 0.924 and 0.912, respectively, which indicated that the identification results are reliable.

**Table 2.** Analysis of water extraction for the downstream watershed of the Minjiang River.

|  | Landsat | Sentinel |
| --- | --- | --- |
| Accuracy | 0.960 | 0.966 |
| Precision | 0.997 | 0.986 |
| Recall | 0.943 | 0.963 |
| F1 | 0.969 | 0.974 |
| Kappa | 0.912 | 0.924 |

### 4.2. Overall Evolutionary Characteristics of the Downstream of Minjiang River

Figure 6 presents the annual variations in runoff and sediment transport at the Zhuqi hydrological station from 1986 to 2021. Over this timeframe, the annual runoff displays a fluctuating pattern, with the highest value in 2016 and the lowest in 1991, without a significant overall trend. There are some fluctuations in the annual sediment transport from 1986 to 1992, which decreases significantly after 1993. Moreover, the relationship between flow and sediment during the periods of 1986–1992 and 1993–2021 is shown in Figure 7. A strong linear relationship is observed between annual runoff and sediment transport from 1986 to 1992, with a linear regression $R^2$ value of 0.985. However, after 1993, this relationship decreases, with an $R^2$ value of 0.544. Additionally, the annual sediment transport shows a more obvious decreasing trend under the same annual runoff conditions.

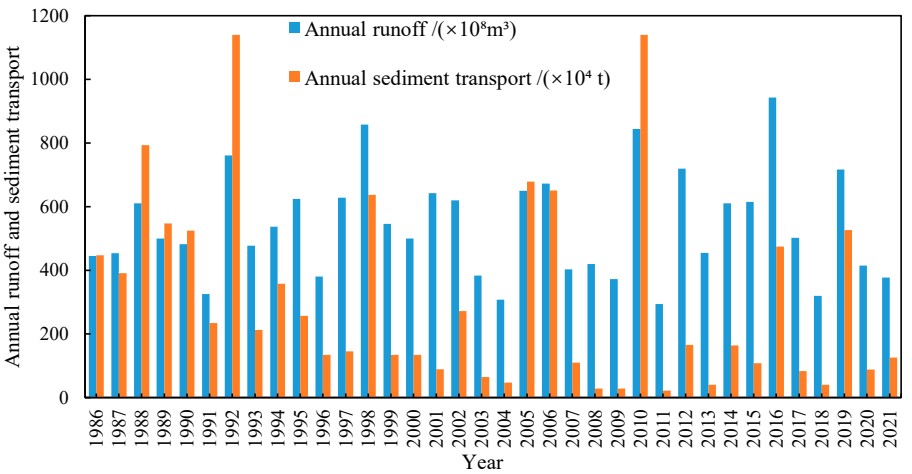

**Figure 6.** Annual runoff and sediment transport at Zhuqi station.

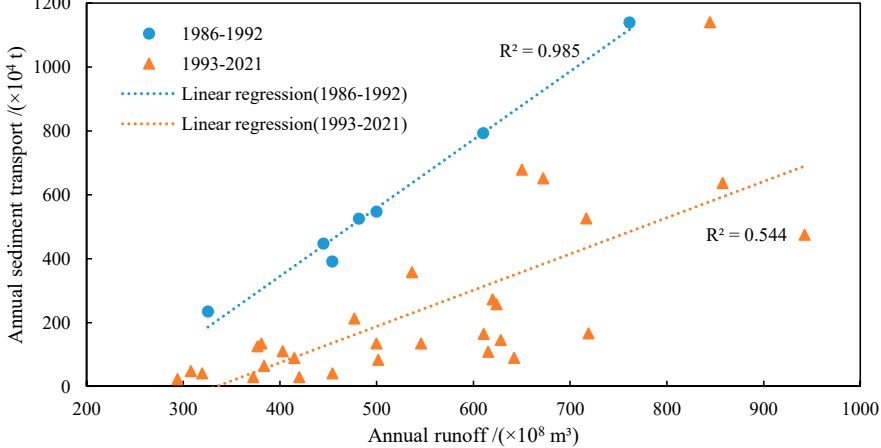

**Figure 7.** Correlation between annual runoff and sediment transport at Zhuqi Station.

Figure 8 shows a distinct trend in the water area of the lower Minjiang River, with an initial increase, followed by a decrease and eventual stabilization. The water area increased during 1986–1997 and experienced a decline during 2000–2003. Since 2003, there has been a tendency toward stabilization in the overall water area, with the declines being comparatively smaller. The sandbar area has remained stable over the last 35 years, with an average annual area of 24.58 km². In contrast, there has been a significant reduction in the beach area, which went from 37.09 km² in 1986 to 16.44 km² in 2021, representing a decline of approximately 20.65 km².

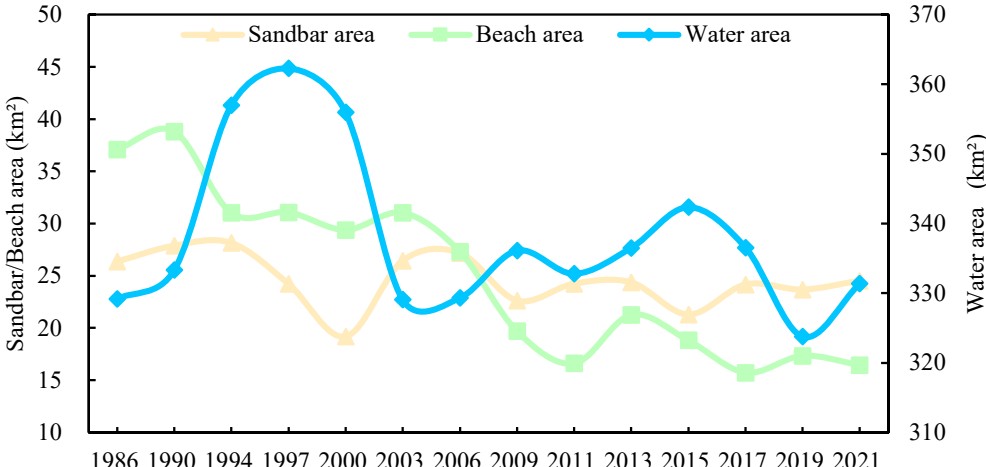

**Figure 8.** The changes in water body, beach, and sandbar areas in the downstream of Minjiang River.

The transformation of land use during 1986–2021 can be observed in Figure 9, where the most significant change takes place in the NG section. Numerous beaches are converted into water bodies in the upper part of the NG section, and the change in river morphology downstream of the NG section is also complex, with major transformations between sandbars, beaches, and water bodies.

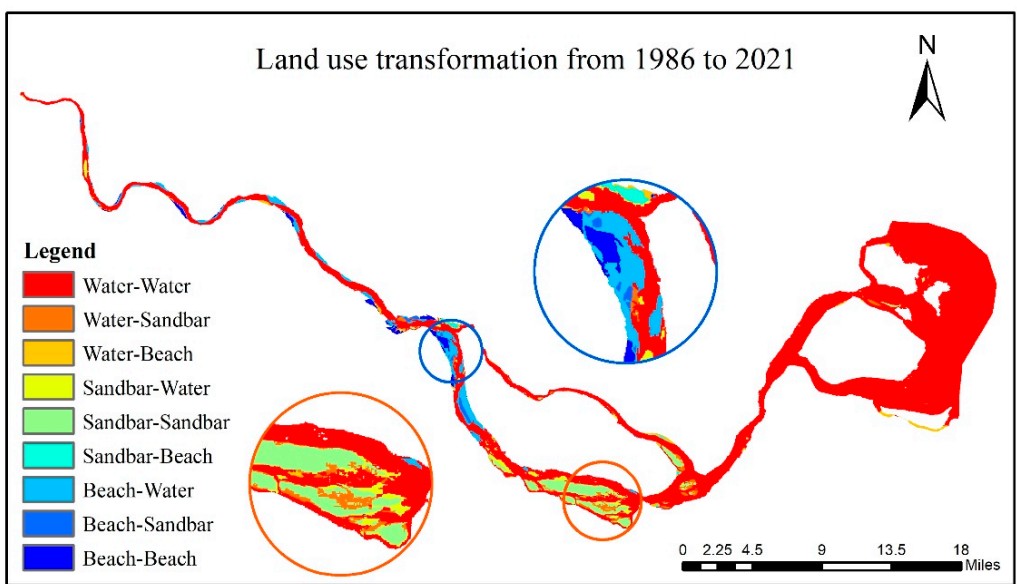

**Figure 9.** The land use transformation from 1986 to 2021 (the blue and orange circled areas represent the northern and southern part of the NG section, respectively).

Between 1990 and 1994, there was a particularly drastic exchange between beaches and water, with approximately 12.25 km² of beaches being converted into water, while the area transformed into sandbars during the same period was only 0.73 km². From

1994 to 2000, 9.90 km$^2$ and 7.40 km$^2$ of sandbars and beaches were converted into water bodies, respectively. However, there was a significantly decreased area of water bodies during 2000–2006, with about 10.35 km$^2$ and 9.95 km$^2$ converted into beaches and sandbars, respectively. Over the period of 2006–2021, there was relatively less variation in land use changes, with the conversion between all types of parcels remaining within 5 km$^2$.

### 4.3. Evolutionary Characteristics of the River Sections

To obtain a more detailed understanding of the changes in the Minjiang River, the study area is divided into four distinct sections, as displayed in Figure 1. These sections include the following: MQMH section, NG sections, BG sections, and MW.

#### 4.3.1. MQMH Section

The primary channel of the MQMH section is narrow and elongated. To comprehensively analyze variations within MQMH, twenty-nine cross-sections (MQ1–MQ29) were established, and the evolution of each cross-section is illustrated in Figure 10. While the upstream segment of MQMH remains relatively stable, with minimal interannual variability in its main channel, there are more significant oscillations in the downstream areas at the river bends. Between 1986 and 2003, the river's course in the MQ25–MQ29 cross-section exhibited significant fluctuations characterized by amplitudes exceeding 100 m, alternating between swings to the left and right banks. Since 2003, the river has been stabilizing, leading to a decrease in the overall amplitude of the river's swing.

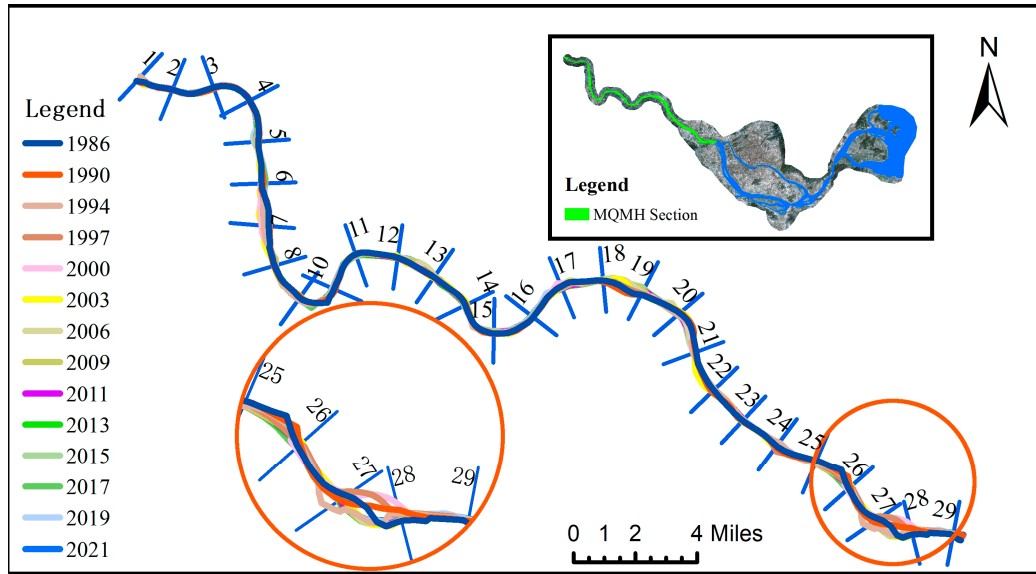

**Figure 10.** Main channel changes in the MQMH section.

Based on the topographical changes observed in the river's cross-section, the upstream river in the MQMH section has been gradually deepening, with fewer changes in the submerged topography of the river occurring after 2016 (Figure 11). Similarly, the downstream channel of the MQMH section (MQ27, MQ28) has also experienced a deepening trend overall, with some oscillation, particularly in MQ28. At the position of the deep flood line, MQ28 oscillated by over 100 m from 2008 to 2016. The degree of change in the river's underwater topography decreased after 2016.

The distribution of river water, sandbars, and beaches in the MQMH section is depicted in Figure 12, showing a significant reduction in the beach area over the last 35 years. Figure 13 depicts a decrease in the prevalence of sandbars throughout the section. During this period, the total water area experiences two significant fluctuations: a gradual decrease between 1986 and 1990, a notable increase from 1990 to 1994, and another gradual decrease. Starting in 2003, the water area began to increase again, stabilizing after 2019. The beach area

shows a marked decline. As of 2021, the beach area had decreased to 5 km$^2$, representing a reduction of around 12.09 km$^2$ from its 1986 level of 17.09 km$^2$.

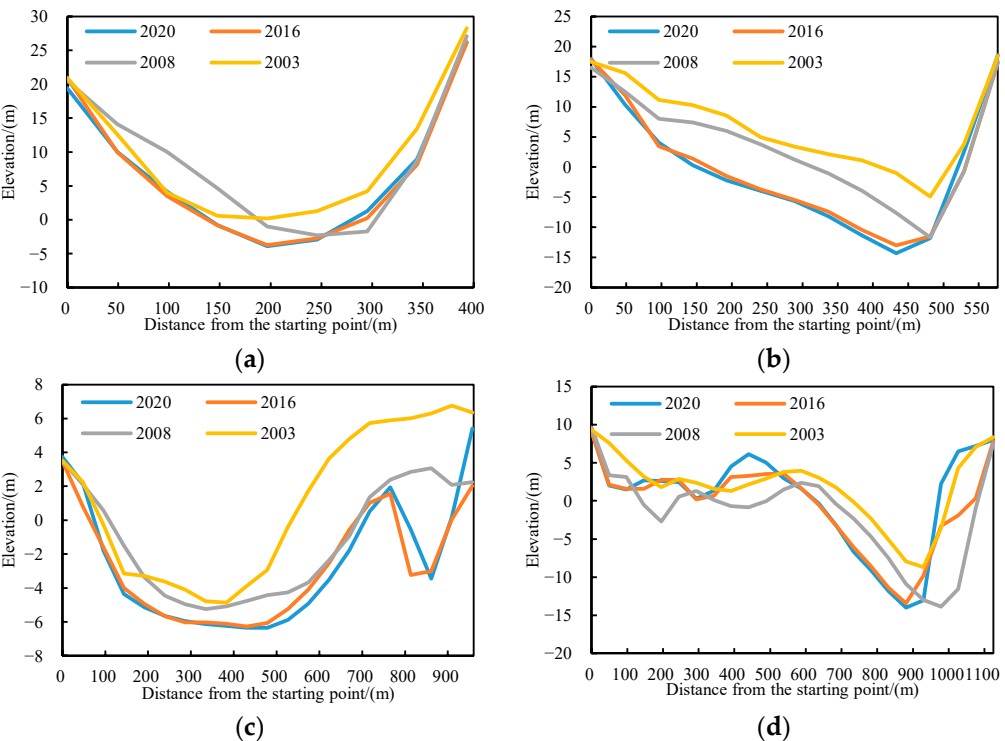

**Figure 11.** Topographical changes in cross-section in MQMH section (the starting point is located on the left bank). (**a**) MQ4; (**b**) MQ10; (**c**) MQ27; (**d**) MQ28.

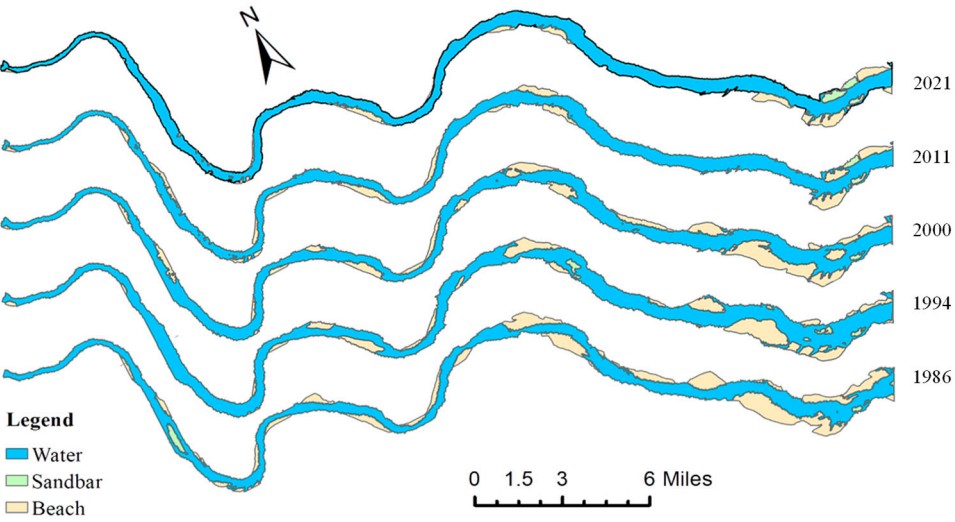

**Figure 12.** Distribution of water, sandbars, and beaches of the river in MQMH section.

The MQMH section of the river channel is moderately curved, with an overall curvature range of 1.25 to 1.29 degrees (Appendix A Figure A1). The curvature increased from 1986 to 2003, reaching a maximum value of 1.29 degrees. However, after 2011, the changes in curvature became less pronounced and stabilize at approximately 1.26 degrees. Compared to the measurements taken in 1986, the channel's curvature has decreased by approximately 0.01.

Fractal dimension calculations reveal that the river channels underwent significant changes before 2003. Specifically, between 1990 and 1994, the fractal dimension of the channel increased sharply from 1.46 to 1.55 degrees, resulting in a curvier overall channel

shape. Subsequently, during 1994–2003, the fractal dimension exhibited an overall decreasing trend, reaching 1.47 degrees by 2003. After 2003, changes in the fractal dimension of the channel became less significant, following a pattern of fluctuating increases before eventually stabilizing.

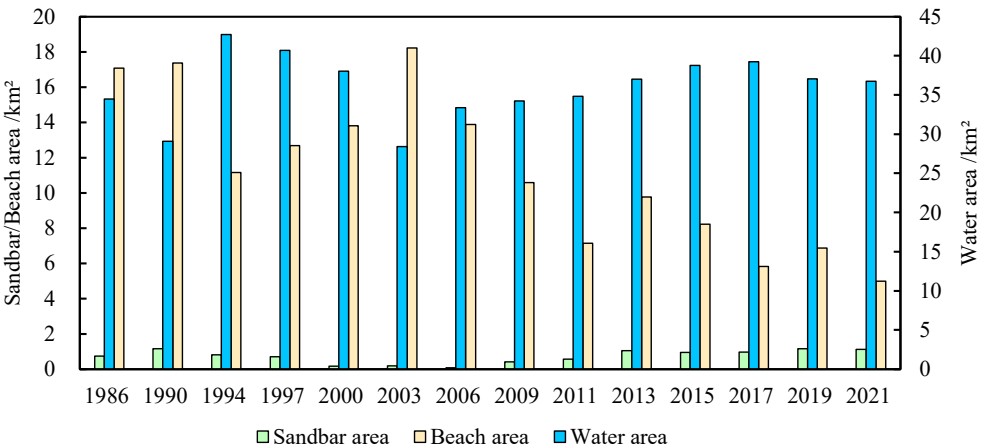

**Figure 13.** Land used transformation of MQMH section from 1986 to 2021.

4.3.2. BG Section

The upper reaches of the BG section are relatively narrow, while the lower reaches exhibit a comparatively wider profile, averaging around 618 m in width over multiple years. Except for the pronounced curvature near the upstream entrance (about 90°), the majority of the river appears relatively straight.

To understand variations in BG section, twenty-eight cross-sections (BG1–BG28) are established, and the corresponding layout and changes in width for each cross-section are illustrated in Figure 14. Differing from the MQMH section, there is an overall decreasing trend in the change in the channel width in the BG section. BG19 experiences the most significant shift in cross-sectional width, with a cumulative decrease of about 378 m over the past 35 years. In contrast, BG4 is observed to be the most stable section, exhibiting a cumulative change of only 259 m over the same period. Ultimately, the river's average width has decreased by approximately 200 m from 1986 to 2021, dropping from the initial measurement of 708 m to 500 m.

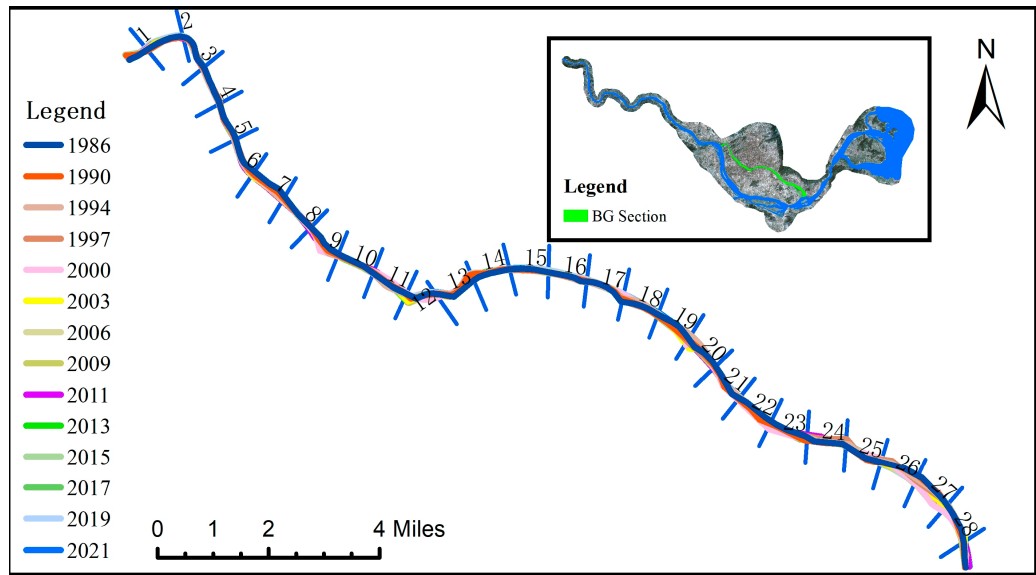

**Figure 14.** Main channel changes in the BG section.

Figure 15 shows the distribution of water, sandbars, and beaches of the river in the BG section, and Figure 16 presents variations in the area of the BG section, revealing that the water area has been declining over time. Although the watershed area increases from 1990 to 1997, it shows a decreasing trend from 1997 to 2011, and the area stabilizes thereafter. By 2021, the overall decrease in the water body area is about 6.73 km$^2$ compared to 1986. The area of the sandbar initially experiences a growth phase from 1986 to 1990, followed by an oscillating downward trend. Until 2015, the magnitude of changes in the area of the sandbars began to decrease, and a certain degree of increased trend appeared. The beach area significantly decreased between 1986 and 1990, partially recovering between 1990 and 1994. Similar to the changes in the sandbar area, it showed a fluctuating downward trend until 2009. After that, the beach area gradually increased and recovered to 1.78 km$^2$ in 2021.

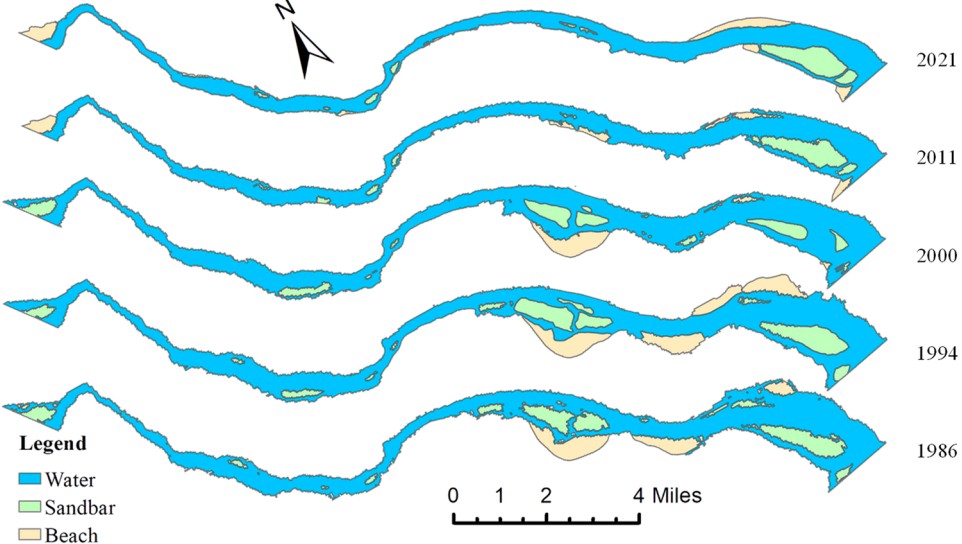

**Figure 15.** Distribution of water, sandbars, and beaches of the river in BG section.

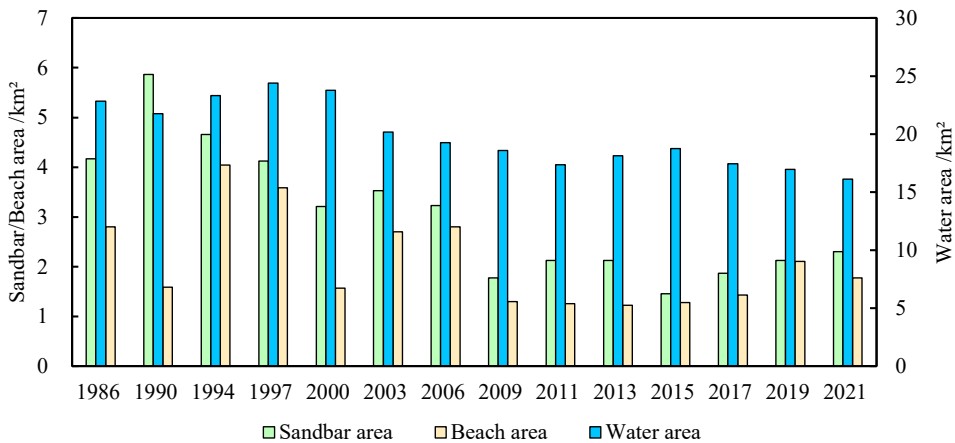

**Figure 16.** Land used transformation of BG section from 1986 to 2021.

The curvature of the BG section has remained relatively stable over the past 35 years, ranging from 1.13 to 1.15, indicating that the river has remained relatively straight (Figure A1). The fractal dimension of the river ranges from 1.47 to 1.54, with an increase of about 0.02 between 1990 and 1997, but it shows an overall decreasing trend. By 2021, the fractal dimension of the river decreased to 1.47, indicating that the overall shape of the river tends to be stable and continuous, and the fragmentation of the river has decreased.

### 4.3.3. NG Section

The remote sensing images show that the NG section has a wider river channel and a complex downstream river morphology.

Using the method of equidistant division, twenty-eight characteristic cross-sections are identified in the NG section (Figure 17). On average, the cross-sections are approximately 1539 m wide, with the widest part of the channel measuring 3400 m at NG17. Compared to the BG section, the NG section displays more significant channel variations. Over the past 35 years, approximately 11 cross-sections have experienced an accumulated change distance exceeding 2500 m, with the greatest change distance recorded at NG2 measuring 3557 m. The river's centerline oscillation is most pronounced at NG5–NG10 in the BG section. Prior to 2009, the river underwent violent oscillations with amplitudes exceeding 100 m.

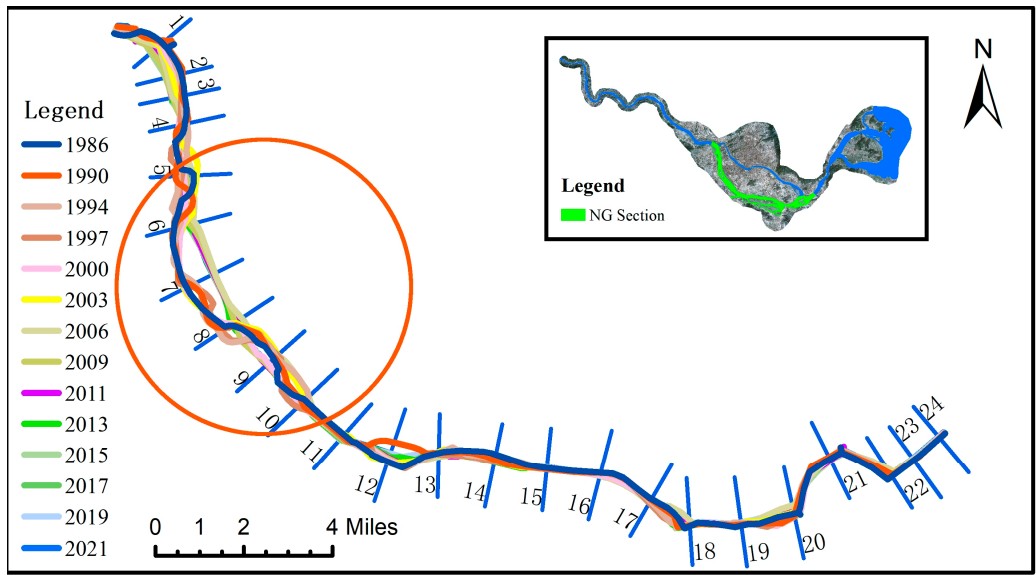

**Figure 17.** Main channel changes in the NG section.

Figure 18 illustrates the topographical changes in some cross-sections in this section. Between 2003 and 2008, the NG2 cross-section experienced significant erosion, resulting in the lowest point of the riverbed reaching an elevation of nearly −20 m. Afterward, from 2008 to 2016, the main channel experienced sedimentation, with a maximum sediment height exceeding 10 m. Similarly, the NG6 and NG9 cross-sections experienced significant erosion between 2003 and 2016, and the depth of the riverbed increased until the river stabilized after 2016, whereas the main channel at NG17 continued to be eroded from 2003 to 2020, while the shoal area showed some sedimentation after 2008.

The distribution of water, sandbars, and beaches of the river in NG section are shown in Figure 19. Between 1986 and 2021, the water surface width showed an oscillating trend of growth. The average cross-sectional width increased by 253 m during 1990–2000. In contrast, the average channel width decreased by 121 m between 2000 and 2011. Thereafter, the rate of channel width change gradually slowed down, with an average increase of about 28 m between 2011 and 2021.

The water area of the NG section underwent a slight decrease from 1986 to 1990 (from 61.14 km$^2$ to 60.49 km$^2$), followed by a significant increase between the early 1990s and 2000 (Figure 20). Subsequently, the water area decreased markedly between 2000 and 2003, and increased dramatically again from 2006 to 2009. During 2013–2021, the average water area remained around 68.05 km$^2$. With respect to the sandbar area, it has remained relatively stable, with fluctuations within 4 km$^2$ over the past 35 years. In terms of the beach area, there was a significant decline from 1990 to 2009, with an overall reduction of roughly 14.90 km$^2$. After this period, the overall beach area was stabilized.

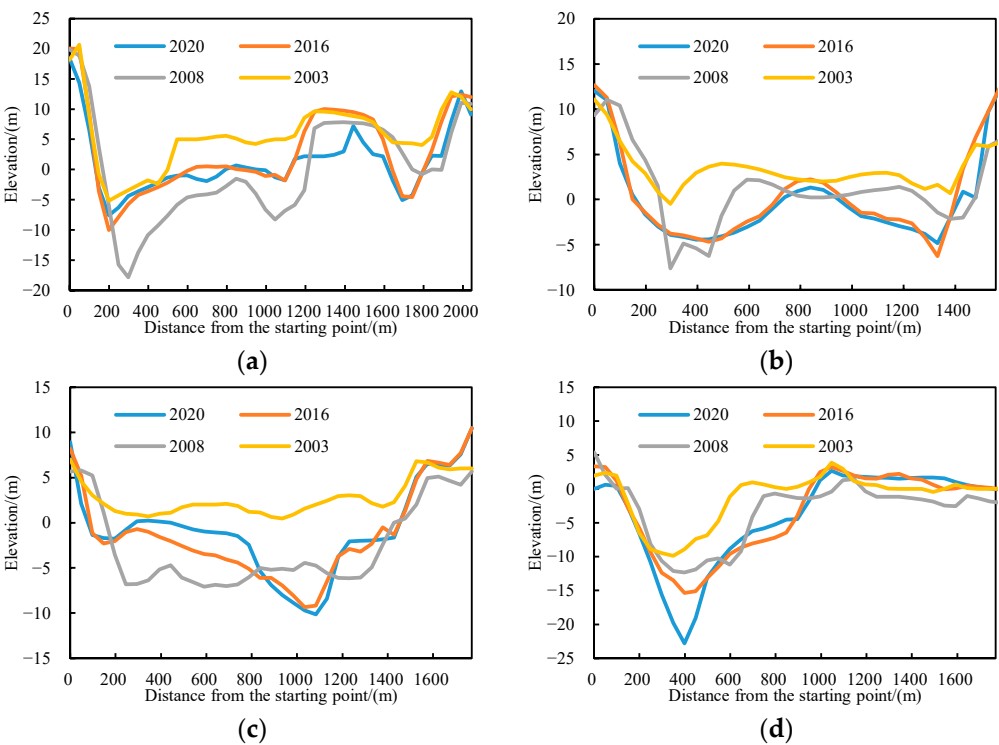

**Figure 18.** Topographical changes in cross-section in NG section (the starting point is located on the left bank). (**a**) NG2; (**b**) NG6; (**c**) NG9; (**d**) NG17.

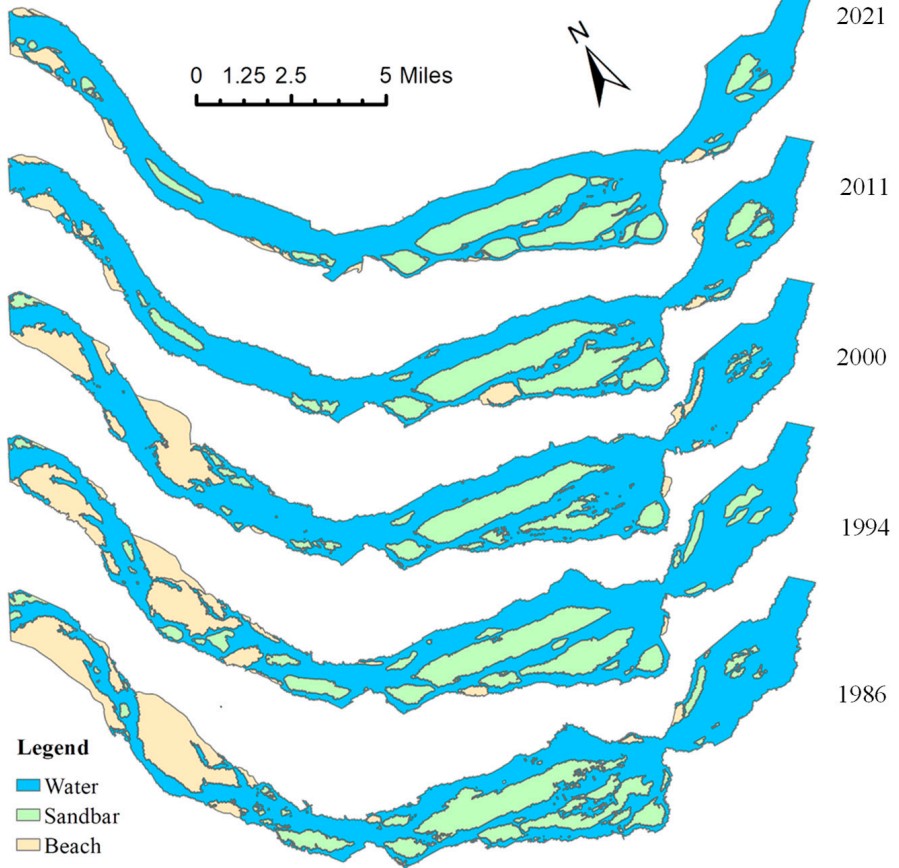

**Figure 19.** Distribution of water, sandbars, and beaches of the river in NG section.

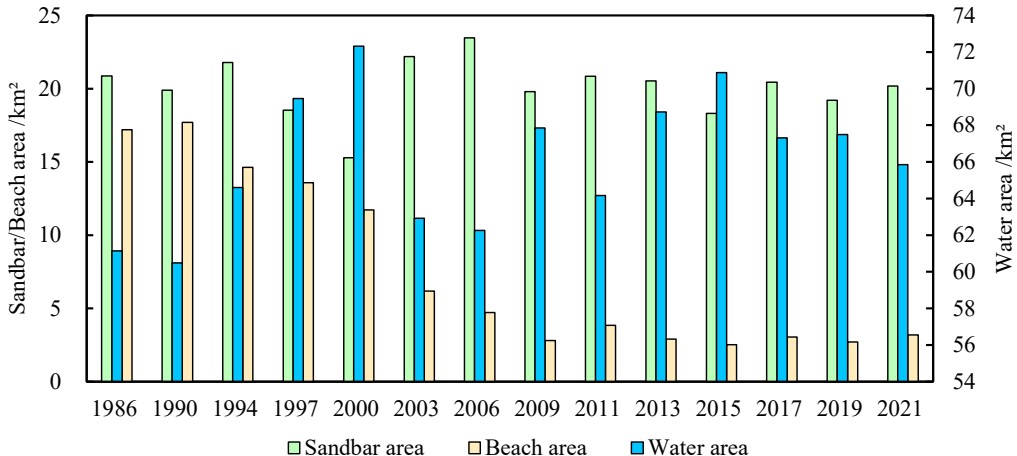

**Figure 20.** Land used transformation of NG section from 1986 to 2021.

The NG section features the most complex channel morphology in the study area. Due to the large number of channel branches in the NG section, only the primary channel curvature was calculated in this study. During the last 35 years, the primary channel curvature changed by about 0.08 (Figure A1), exhibiting an overall oscillating downward trend (from 1.38 in 1986 to 1.30 in 2021). After 2006, the variation in channel curvature decreased, with an overall average of about 1.31. In general, the fractal dimension of the river shows a trend of increasing, decreasing, and finally stabilizing.

### 4.3.4. MW Section

The MW section is formed by the confluence of the NG and BG sections, featuring a wide channel, slow flow, and direct connection to the sea, with the characteristics of a typical estuarine channel.

To analyze the changes in the cross-section width of the MW section, 29 cross-sections (MW1–MW29) are established, covering the Meihua Channel and Guantou Channel (Figure 21). The morphology of the MW section of the river is relatively stable, but in recent years, the width of the river has shown a slight decreasing trend (Figure 22). In the past 35 years, the average width of the cross-section has decreased by about 121 m, with an average contraction rate of 3.46 m/a.

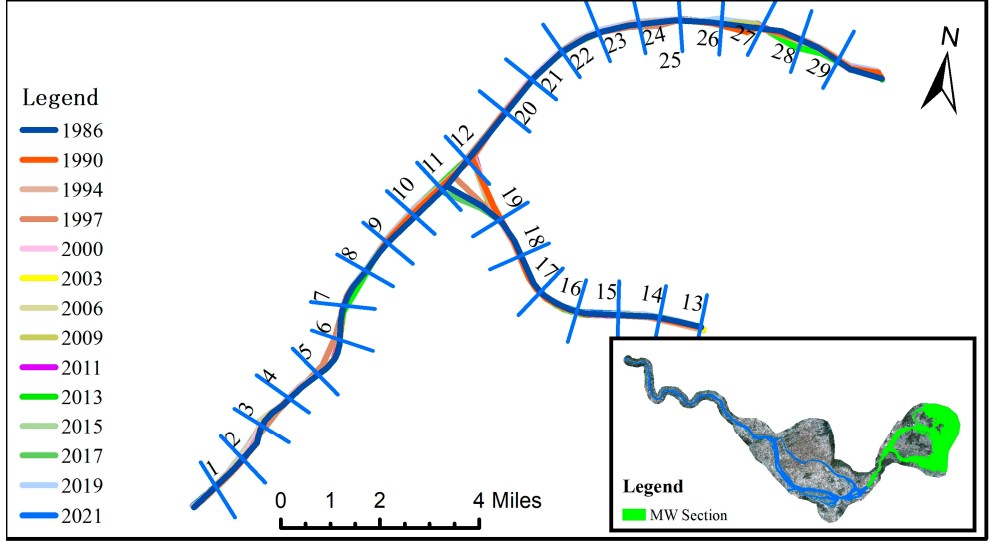

**Figure 21.** Main channel changes in the MW section.

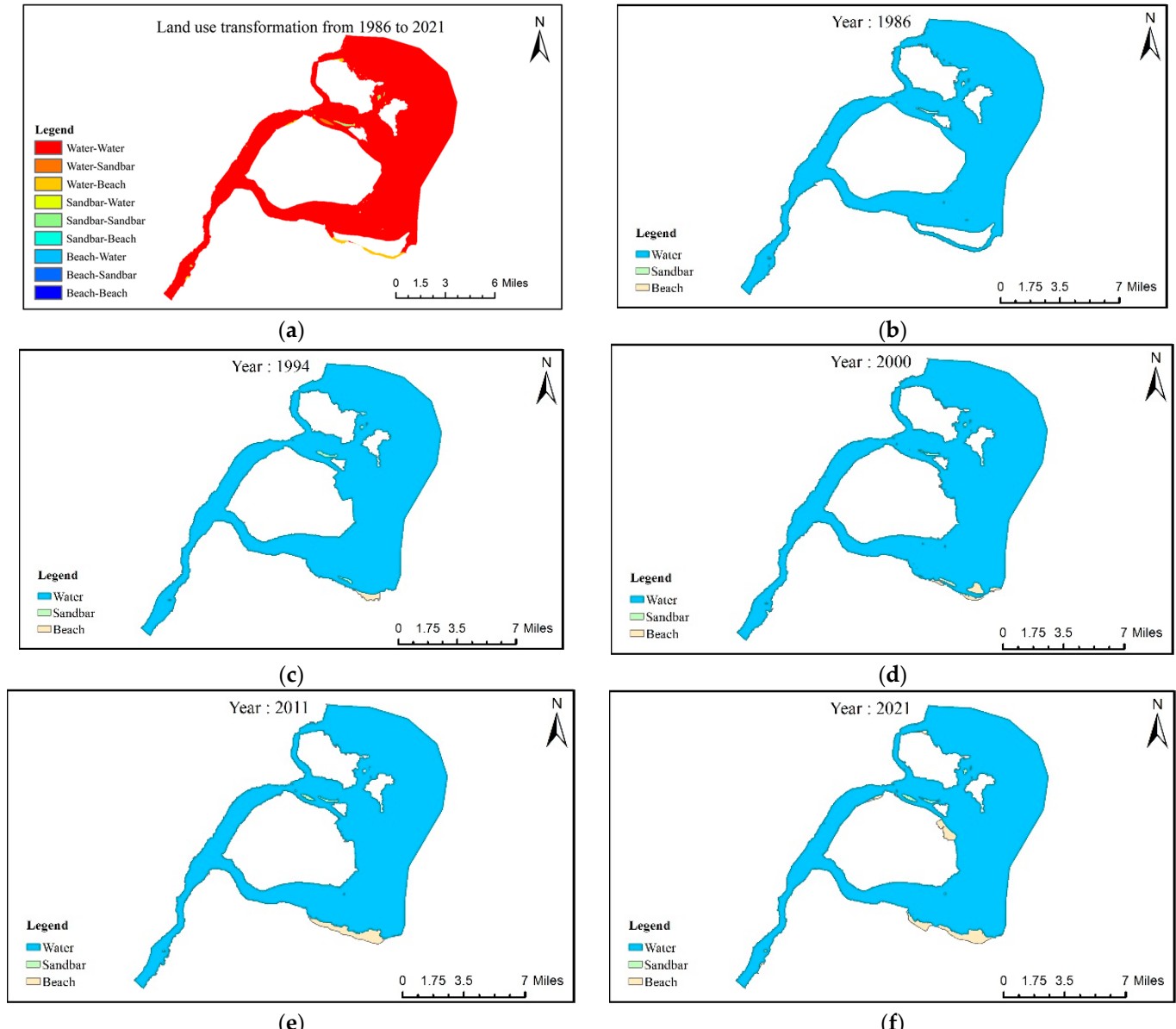

**Figure 22.** Distribution of water, sandbars and beaches of the river in MW section. (**a**) Land use transformation from 1986 to 2021; (**b**) 1986; (**c**) 1994; (**d**) 2000; (**e**) 2011; (**f**) 2021.

Regarding the water area, the MW section exhibits a trend of first increasing and then decreasing as shown in Figure 23. During 1986–1997, the water area increased by about 16.99 km$^2$ and then showed a gradual decrease. Unlike other river sections, the area of the marginal beach in the MW section showed an increasing trend, increasing by about 6.47 km$^2$ between 1986 and 2021. The area of the sandbar stays below 1.50 km$^2$ (excluding the larger islands), and the overall change is small.

The overall morphology of the MW section is relatively stable, with the curvature of the primary channel always ranging from 1.22–1.25 and the fractal dimension ranging from 1.80–1.82, with small overall change trends and relatively stable channel (Figure A1). The reason for the larger fractal dimension in this section is that some islands are included in the calculation process, and the shoreline is longer and more complex, which makes the fractal dimension calculation larger.

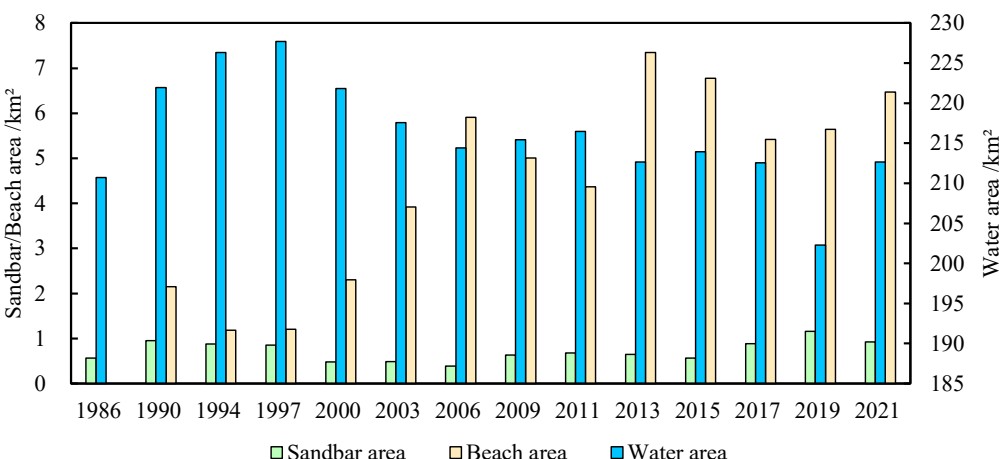

**Figure 23.** Land used transformation of MW section from 1986 to 2021.

## 5. Discussion

### 5.1. Impact of Large Hydraulic Projects on River Morphology

Large hydraulic projects, such as the construction of reservoirs, have significant impacts on river morphology, resulting in both vertical and horizontal changes in downstream rivers [37]. This includes bed scouring, bed sand coarsening, longitudinal ratio drop adjustment, mainstream oscillation, river width scale adjustment, and river curvature adjustment. These changes occur gradually over time, ultimately resulting in a new riverbed morphology that is adapted to the altered water and sand conditions.

The construction of the Shuikou Reservoir in 1993 in the lower reaches of the Minjiang River provides a typical example of these effects, which resulted in significant changes to the river's morphology. Due to its regulation function, the discharge flow during non-flood periods is guaranteed, leading to a gradual increase in the water area of each section in April from 1900 to 1994.

In addition, due to the reservoir's sediment retention function, the sediment content in the inflow of the downstream river channel significantly decreases. Except for the years when large-scale flood events occurred, in 1998, 2005, 2006, and 2010, the sediment transport in other years significantly decreased compared to the pre-construction period. Among them, the sediment transports in 2008, 2009, and 2011 were as low as 288,000 tons, 287,000 tons, and 222,000 tons, respectively (Figure 6). Furthermore, the erosion of sandbars and beaches intensified under the influence of clear water discharge, and the area continued to decrease. From 1990 to 1994, the beach area decreased by 7.8 km$^2$, accounting for about 20% of the same period's beach area. The foundation scouring of river-related engineering is severe, and collapse accidents occur frequently. Therefore, the construction of Shuikou Reservoir plays a significant role in the evolution of river morphology in the lower reaches of the Minjiang River.

### 5.2. Impact of River Sediment Mining on the Evolution of River Morphology

River sediment mining also has a significant impact on the evolution of river morphology in the lower reaches of the Minjiang River. The large-scale excavation of river sediment mining began in the early 1990s and resulted in numerous problems, including the collapse of the Jiefang Bridge, flood embankment breaches, and coastal road subsidence. Before 1997, sediment mining activities were mainly concentrated in the BG section, and the excessive sand mining caused a continuous decrease in the river bed level, which worsened the upstream migration of saltwater during the dry season and negatively impacted water supply security.

Since the ban on sediment mining in BG in 1997, sediment mining activities have been mainly concentrated in the NG section. According to the sediment mining plan in recent years, the sediment mining volume in NG was 4.55 million m$^3$ (including 3.75 million m$^3$

of dredging) in 2010; 600,000 m$^3$ in 2011; 630,000 m$^3$ in 2012; 600,000 m$^3$ in 2013; and 600,000 m$^3$ in 2014. This directly led to a significant reduction in the beach area in the NG section, with a decrease of about 11.02 km$^2$ between 1997 and 2015. The trend of decreasing beach areas was curbed after the ban on sediment mining in NG in 2015. Coupled with the construction of spur dikes and other engineering measures, the flow rate on both sides of the river slowed down, and the beaches were recreated. The impacts of river sediment mining on the evolution of river morphology are complex and multi-faceted, and it is necessary to continue to monitor and regulate sediment mining activities to ensure the sustainable development of the river's ecosystem.

*5.3. Impact of River Training Measures along the River*

River training measures are human interventions that aim to modify the natural course of a river to achieve specific goals, such as flood control, navigation, and water supply. These measures can have significant impacts on the river's morphology, altering its width, depth, and curvature. In the case of the Minjiang River, river training measures have been implemented to address issues such as river scouring and poor scouring resistance of riverbanks.

From 1990 to 2009, the NG section experienced scouring, leading to a decrease in the area of beaches and a reduction in navigability. To address this issue, Fuzhou City launched the NG channel improvement project in 2010, which involved dredging, dam throwing, and beaconing. The project dredged a total of 4,564,500 m$^3$, and some spur dikes were laid along the improvement section to adjust the flow direction (Figure A2). These river training measures have significantly contributed to stabilizing the river's morphology and enhancing its flood control capacity. As a result of the construction of these measures, the trend of decreasing beach and sandbar areas in the NG section has been mitigated, leading to a reduction in flow velocity on both sides of the river and sediment accumulation, thereby increasing the area of beaches and sandbars. The primary channel's curvature also decreased from 1.32 to 1.30 between 2009 and 2011. Consequently, the water level in the river's center rose, improving its navigability.

*5.4. Influence of Geological Conditions on River Morphology*

Geological conditions also have an important influence on the evolution of river morphology. After analyzing remote sensing images, it is evident that the downstream of the MQMH has experienced the most significant changes, particularly in the central line of the primary channel. In contrast, the upstream section has maintained a more stable morphology with minimal planar changes, which can be attributed to the geological conditions of the river.

Geological exploration results suggest that the upstream section of MQMH flows through mountainous regions containing many canyons where both banks are primarily composed of granite. Additionally, the upper layers consist mainly of medium sand and pebbles [38], with medium sand thickness ranging between 2.65 and 11.73 m, pebble thickness between 4.25 and 6.40 m, and a -drop of approximately 0.08‰. On the other hand, the downstream section features a hilly and plain landscape with a broadened river channel and a slower flow rate. The riverbed is generally shallow, with varying widths and many sandy beaches, composed of clay and sand that are poorly resistant to erosion and prone to collapse. These geological conditions have contributed to the upstream section maintaining a more stable morphology, while the downstream section has experienced dramatic changes in width and oscillation due to poor resistance to erosion by floodplains.

The Fujian government has carried out a large amount of river channel management work in the downstream of the MQMH to mitigate the problem of large swings and instability in the river channel. A total of 16.8 km of embankments have been built to provide certain control and reduce the swing amplitude of the main river channel. After 2003, the river oscillation problem was controlled to some extent.

*5.5. Impact of Large Flood Events on River Morphology*

Large flood events have a significant impact on the morphology of rivers, particularly in downstream areas. The Minjiang River basin has experienced several large flood events in recent years, including the 100-year flood in 22–24 June 1998, which had a maximum flow of 33,800 m$^3$/s, measured at the Zhuqi hydrological station. Subsequent large floods occurred in 2002, 2005, 2006, and 2010.

Although flooding does not directly change the planform of the primary channel of the lower Minjiang River in a short period of time, it has caused significant channel incisions, resulting in an obvious change in the ratio of river diversion [39]. Since 2002, the diversion ratio of BG has continuously decreased (Table A2). In September 2002, the average diversion ratio of NG and BG was 21.47% and 78.53%, respectively, at a flow rate of 1500–2000 m$^3$/s at Zhuqi hydrological station. While in February 2008, the average diversion ratio of NG and BG was 83.23% and 16.77%, respectively, at the same flow rate at Houguan hydrological station. This reveals that the diversion ratio of NG and BG underwent a significant change during the non-flood season, with the diversion ratio of NG increasing significantly, now reaching about 80%. Since 2003, NG has been continuously deepening, particularly in the main channel, while BG has also experienced erosion, albeit to a lesser extent compared to NG. As a result, the BG channel has lost its previous narrow and deep pattern, while the NG channel has become both wide and deep.

The change in diversion ratio will in turn affect the adjustment of erosion and sedimentation patterns in the riverbed and other corresponding changes. The increase in water flow from the NG during the dry season will inevitably increase the erosion of sandbars and beaches, whereas the decreased water flow from the BG will cause a significant drop in the river level, affecting water intake safety. Thus, the long-term impacts of major flood events on river morphology should be considered in river management and planning decisions.

## 6. Conclusions

This study proposes an integrated approach utilizing remote sensing image data combined with deep learning and visual interpretation algorithms to analyze the river's morphological evolution. The research focuses on the lower reaches of the Minjiang River in China and comprehensively analyzes the river's morphological evolution from 1986 to 2021. The study leads to the following conclusions:

(1) The proposed method of river water identification in this study demonstrates high accuracy and effectiveness, with an F1 score and Kappa coefficient greater than 0.96 and 0.91, respectively.

(2) By integrating the fractal dimension, river curvature, and land use transfer matrix, the plane morphological evolution of the river can be comprehensively characterized. The results reveal that the downstream of the MQMH section experienced beach erosion, while the upstream section maintained a more consistent morphology. The NG section widened, and the BG section contracted. The changes in river morphology at the MW section were relatively minor.

(3) The morphological evolution of the lower reaches of the Minjiang River has been significantly impacted by various factors, including reservoir construction, river sediment mining, river training measures, geological conditions, and large flood events.

The consequences of channel incisions have intensified the effects of tidal action downstream of the Minjiang River. With this in mind, we will continue monitoring its evolution while exploring how tidal action and human engineering measures act jointly on the river's morphological changes.

**Author Contributions:** Conceptualization, N.Z.; Funding acquisition, H.C.; Investigation, S.S. and L.-Y.H.; Methodology, N.Z.; Resources, L.-Y.H. and B.-R.T.; Supervision, H.C.; Writing—review and editing, N.Z., S.S., B.-R.T., H.C. and C.-Y.X. All authors have read and agreed to the published version of the manuscript.

**Funding:** This research was funded by the National Key Research and Development Program grant number (2022YFC3002701) and Water Science and Technology Project in Fujian Province, China.

**Data Availability Statement:** The annual runoff and sediment transport data of Zhuqi hydrological station can be found in http://www.mwr.gov.cn/sj/tjgb/zghlnsgb/ (accessed on 27 April 2023).

**Acknowledgments:** This research was supported by the National Key Research and Development Program (2022YFC3002701) and Water Science and Technology Project in Fujian Province, China. The authors thank the project for supporting the study.

**Conflicts of Interest:** The authors declare that they have no known competing financial interests or personal relationships that could have appeared to influence the work reported in this paper.

## Appendix A

**Table A1.** Statistical table of land type transformation.

| Year | | Area (km²) | | | | | | |
|---|---|---|---|---|---|---|---|---|
| | | **1986–1990** | **1990–1994** | **1994–2000** | **2000–2006** | **2006–2011** | **2011–2017** | **2017–2021** |
| Transform type | Sandbar–Water | 5.12 | 5.57 | 9.90 | 0.99 | 4.01 | 3.15 | 2.85 |
| | Beach–Water | 3.09 | 12.25 | 7.40 | 5.94 | 4.80 | 3.30 | 1.25 |
| | Water–Water | 307.65 | 321.63 | 335.67 | 321.10 | 319.50 | 324.00 | 319.87 |
| | Sandbar–Beach | 0.18 | 0.85 | 1.21 | 1.26 | 1.20 | 0.00 | 0.05 |
| | Beach–Beach | 28.69 | 23.50 | 19.15 | 14.05 | 13.57 | 9.70 | 10.72 |
| | Water–Beach | 6.65 | 2.91 | 7.52 | 10.35 | 1.19 | 2.20 | 3.37 |
| | Sandbar–Sandbar | 21.00 | 21.42 | 16.68 | 15.69 | 20.89 | 20.99 | 21.21 |
| | Beach–Sandbar | 1.05 | 0.73 | 0.74 | 1.33 | 0.42 | 1.58 | 0.01 |
| | Water–Sandbar | 4.96 | 5.90 | 1.62 | 9.95 | 2.71 | 1.41 | 3.27 |

**Table A2.** The average flow diversion ration between NG and BG.

| Station | Time | Flow/(m³/s) | Average Flow Diversion Ratio/(%) | |
|---|---|---|---|---|
| | | | **NG** | **BG** |
| Zhuqi | September 2002 | 1200–1500 | 19.63 | 80.37 |
| | | 1500–2000 | 21.47 | 78.53 |
| | | 2000–2600 | 30.94 | 69.06 |
| | February 2008 | 1500–2000 | 83.23 | 16.77 |
| | | 2000–3000 | 65.18 | 34.82 |
| Houguan | | 3400–4000 | 79.41 | 20.59 |
| | June 2013 | 4000–4500 | 80.05 | 19.95 |
| | | 4500–5300 | 77.80 | 22.20 |

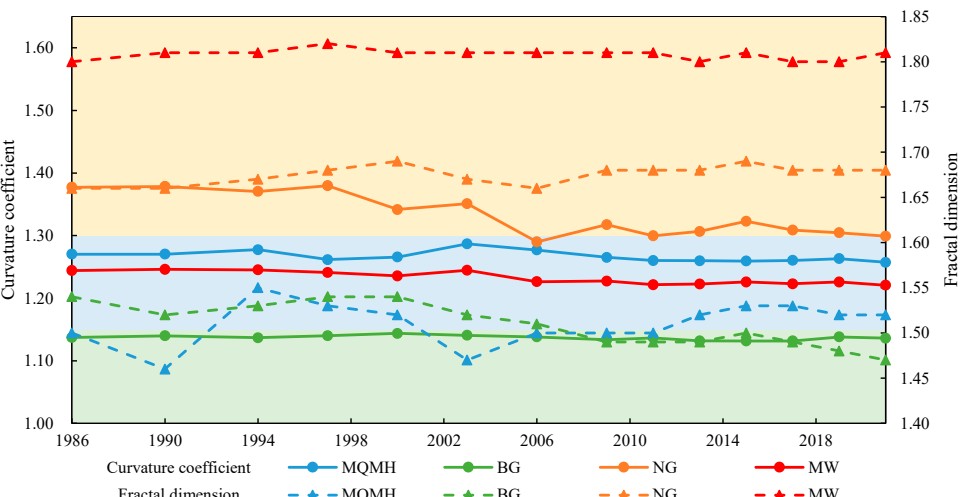

**Figure A1.** The changes in river curvature and fractal dimension.

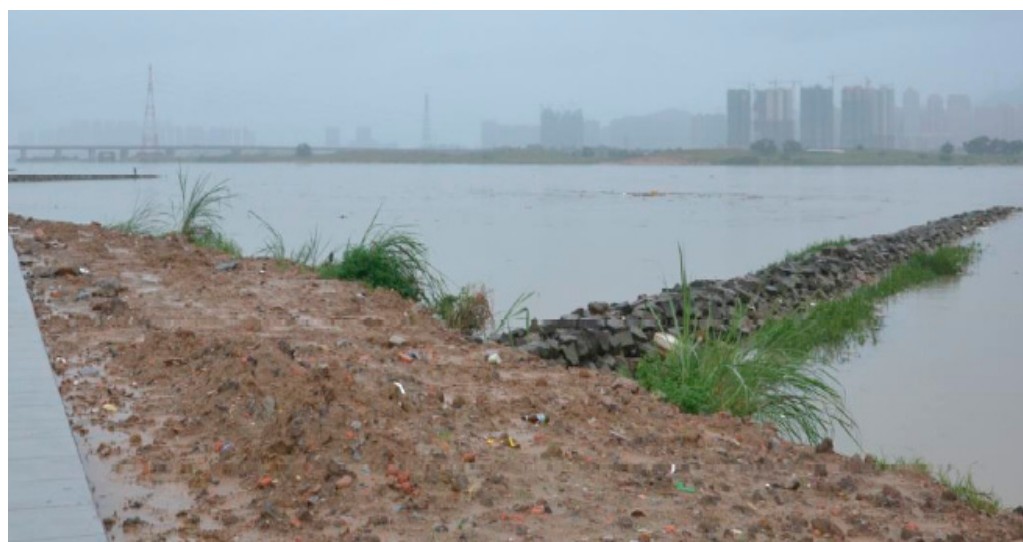

**Figure A2.** Spur dike in NG section.

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
