# Peer review of "An Integrated Approach for Analyzing the Morphological Evolution of the Lower Reaches of the Minjiang River Based on Long-Term Remote Sensing Data"

_remotesensing, doi:10.3390/rs15123093_

Round 1
Reviewer 1 Report
I guess the distance between a river bank against its opposite bank is fixed. An increaed water body area within the banked area NG might not indicate the river width was expanded. Please confirm it.
Here the paper focuss on illustrating the change of the river width when talking aout the morphology of a reach section of the river. It may be more informative to showing the profile of a crosssection and the change of elevation over the years in some interesting areas, i.e. figure 7 (27,28).
Are there any human activities also affect the change of surface water expansion on the downstream river? for exsample, retaining dam.
Author Response
First of all, the authors thank the reviewer very much for the very careful and thorough check of the manuscript. We worked on your corrections, comments and suggestions, and we have addressed most of them in an acceptable way. According to the comments, our replies are listed as following:
Point 1:
I guess the distance between a river bank against its opposite bank is fixed. An increased water body area within the banked area NG might not indicate the river width was expanded. Please confirm it.
Response 1:
Thanks to the reviewer pointing out the mistake of author's expression. Specifically, the expansion of water area within the river banks does not necessarily imply the widening of the river channel. The expression of "the river width shows an oscillating trend of growth" in the manuscript refers to the expanding trend of water surface width under the same remote sensing imaging conditions. The author has revised the relevant expressions in the article to ensure a more rigorous and accurate expression of the viewpoint. The revised sentence is shown as follow:
“Between 1986 and 2021, the water surface width shows an oscillating trend of growth. The average cross-sectional width increases by 253 m during 1990-2000.”
Point 2:
Here the paper focuses on illustrating the change of the river width when talking about the morphology of a reach section of the river. It may be more informative to showing the profile of a cross section and the change of elevation over the years in some interesting areas, i.e. figure 7 (27,28).
Response 2:
Thanks to the reviewer's suggestions. The authors agree with reviewer’s viewpoint that effective cross-sectional elevation information can provide more intuitive information for riverbed erosion and sedimentation. We obtained underwater terrain data for the MQMH and NG sections in 2003, 2008, 2016, and 2020 by applying to the Fujian Provincial Hydraulic Research Institute. Unfortunately, we were unable to obtain data for the BG and MW sections. We have supplemented the longitudinal evolution analysis of some cross-sections to enhance the comprehensiveness of the article. Some of the modifications are shown as below:
Section 4.3.1:
Based on the topographical changes observed in the river cross section, the upstream river in the MQMH section has been gradually deepening, with less change in the submerged topography of the river occurring after 2016(Figure 11). Similarly, the downstream channel of the MQMH section (MQ27, MQ28) has also experienced a deepening trend overall, with some oscillation, particularly in MQ28. At the position of the deep flood line, MQ28 oscillated by over 100 meters from 2008 to 2016. The degree of change in the river's underwater topography decreased after 2016.
Section 4.3.3:
Figure 18 illustrates the topographical changes of some cross-sections in this section. Between 2003 and 2008, the NG2 cross-section experiences significant erosion, resulting in the lowest point of the riverbed reaching an elevation of nearly -20m. Afterward, from 2008 to 2016, the main channel experiences sedimentation, with a maximum sediment height exceeding 10m. Similarly, the NG6 and NG9 cross-sections experiences significant erosion between 2003 and 2016, and the depth of the riverbed increases until the river stabilized after 2016. Whereas, the main channel at NG17 continued to be eroded from 2003 to 2020, while the shoal area showed some sedimentation after 2008.
Point 3:
Are there any human activities also affect the change of surface water expansion on the downstream river? for example, retaining dam.
Response 3:
Thanks to the comment of reviewer. Human activities such as the construction of retaining dams can also affect the change of surface water expansion on downstream rivers. Retaining dams can alter the flow of water downstream by regulating the amount of water released from the dam, which can impact the water surface width and the overall morphology of the river. Specifically, this study discusses the impact of some human activities on the evolution of river morphology in the lower reaches of Minjiang River as well, in sections 5.1-5.3. The construction of the Shuikou Reservoirs (Section 5.1) has caused clear water discharge and exacerbated the longitudinal erosion of the river channel. Excessive sediment mining in the 1990s (Section 5.2) has resulted in riverbed incision and accidents such as the collapse of the Jiefang Bridge. The river training measures in the NG section (Section 5.3) has involved the dredging of the river channel and the construction of some spur dikes, which has significantly contributed to stabilizing the river's morphology and enhancing its flood control capacity.

Reviewer 2 Report
Keywords: The two keywords “river morphological evolution” and “remote sensing” are also present in the title. Please choose two different keywords, thank you.
1. Introduction (pag. 1): “The launch of satellites such as MODIS…” Unlike the Landsat constellation and Sentinel-2 the MODIS has a coarser spatial resolution (250m), so I would not mention it in this sentence.
1. Introduction (pag. 1): “With resolutions up to 1-2 m and sub-meter” Which sensor are you referring to?
2.1 Study area (pag. 2): “Covering an area of 11,575 km², the region had an estimated population of 7.34 million in 2019” Are you referring to the River Basin area?
2.1 Study area (pag. 2): The MQMH section, which is depicted in Fig. 1, is not mentioned in the text.
Fig. 1: The colours of the NG and MW sections are too similar. I would recommend changing the tone. I would also recommend including a small map of China to better understand the location of the river.
Table 1: The capital letter in “Sentinel2” is missing.
2.2.2 Model training dataset (pag. 4): “To improve the model’s recognition ability under diverse conditions, we expand the dataset using image enhancement techniques such as random rotation, contrast adjustment, horizontal and vertical flipping, and brightness contrast adjustment” Has anyone already used this method to expand the dataset? (If so, please insert some citations).
3.1 Remote sensing image pre-processing (pag. 4): “The pre-processing of remote sensing images includes several steps. First, area network leveling is performed using the panchromatic image to match connection points and ensure an error of less than 3 pixels” Please specify that the panchromatic band was only used for Landsat-8 images.
3.1 Remote sensing image pre-processing (pag. 4): Please explain in the text whether you resampled the sensor bands to the same spatial resolution and if you used the images as level 2 (already atmospherically corrected).
3.2 River water interpretation method (pag. 4): “Firstly, neural network is used for automatic recognition of water bodies in the whole area” I do not understand whether the neural networks rely on the spectral signature to perform the interpretation of the image or whether it is a kind of automatic interpretative visualisation.
3.3 Beaches and sandbars interpretation method (pag. 6): “Landsat 5 and Landsat 8 images are then extracted using a combination of bands 543, while Sentinel remote sensing images are identified using bands 654” I recommend rewriting the sentence so that it is better understood which bands have been chosen in the RGB display, for each sensor.
3.5.3 Land use transfer matrix (pag. 7): “As a result, the land use transfer matrix is widely used in analyzing land use change” Please insert some references.
Fig. 5: Missing capital letter in 'water area' in the y-axis.
Fig. 6: Please explain in the caption that the two circled areas represent the northern and southern part of the NG section.
Tab. 3: I would suggest moving this table to the supplementary materials.
Fig. 7: I would suggest inserting a map of the entire river to highlight the location of the MQMH section. This also applies to the other sections of the river (Fig. 11, 14, 17). I would also suggest moving the number of cross-sections towards the extremities so that they are more readable (also in the other figures).
4.3.1 MQMH section (pag. 10): “The distribution of river water, sandbars, and beaches in the MQMH section is depicted in Figure 8 illustrates the distribution of river water, sandbars, and beaches in the MQMH section” Please rewrite this sentence (there are repetitions).
Fig. 9: I would suggest using the same three colours you used for Fig. 8 (water=blue; sandbar=green; beach=yellow). This also applies to the other sections of the river (Fig. 13, 16, 19).
Fig. 10: I would suggest moving this figure to the supplementary materials.
5.3 Impact of river training measures along the river (pag. 19): “The project dredges a total of 4,564,500 m3, and some spur dikes are laid along the improvement section to adjust the flow direction (Figure 20).” Perhaps you meant Fig. 21.
Fig. 21: I would suggest moving this figure to the supplementary materials..
Tab. 4: I would suggest moving this table to the supplementary materials.
Author Response
Responses to the reviewer 2
First of all, the authors thank the reviewer very much for the very careful and thorough check of the manuscript. We worked on your corrections, comments and suggestions, and we have addressed most of them in an acceptable way. According to the comments, our replies are listed as following:
Point 1:
Keywords: The two keywords “river morphological evolution” and “remote sensing” are also present in the title. Please choose two different keywords, thank you.
Response 1:
Thanks to the reviewer's suggestion, the authors have revised the keyword to: River morphology; water body identification; UNet; MobileNet; Minjiang River
Point 2:
- Introduction (pag. 1): “The launch of satellites such as MODIS…” Unlike the Landsat constellation and Sentinel-2 the MODIS has a coarser spatial resolution (250m), so I would not mention it in this sentence.
Response 2:
Thanks to the reviewer’s advice, the authors have revised it in the manuscript.
Point 3:
- Introduction (pag. 1): “With resolutions up to 1-2 m and sub-meter” Which sensor are you referring to?
Response 3:
Thanks to the reviewer for bringing up this issue. The authors' statement are not precise enough. The authors are referring to satellites such as SkySat-1, SkySat-2, and SuperView-1 when discussing local resolution of 1-2m and sub-meter resolution. Specifically, the ground sampling distance (GSD) at the nadir of SkySat-1 and SkySat-2 is 0.86 m for panchromatic bands and 1.1 m for multispectral bands (Aati and Avouac, 2020), while the SuperView-1 satellite captures images with a 0.5 m resolution for the panchromatic channel, and 2 m resolution for the blue, green, red, and NIR multispectral channels, respectively (Liu et al., 2020). The authors have revised the paragraph and added corresponding references to the manuscript. The revised paragraph is shown as follow:
“The launch of satellites, such as Landsat, Sentinel, SkySat, and SuperView, has significantly improved the spatial resolution of remote sensing imagery (Wang et al., 2022). Specifically, the SkySat-1 and SkySat-2 have a GSD of 0.86 m for panchromatic bands and 1.1 m for multispectral bands, while the SuperView-1 captures images with a 0.5 m resolution for the panchromatic channel, and 2 m resolution for the blue, green, red, and NIR multispectral channels (Aati and Avouac, 2020; Liu et al., 2020). Unfortunately, these satellites' data is not currently available for free. On the other hand, the Landsat and Sentinel satellite series provide high-resolution imagery data with spatial resolutions of 30m and 10m, respectively, which offer detailed topographical, vegetative, and hydrological information (Behera et al., 2018). Furthermore, these data are available to the public for free, making them more suitable for broad scientific research. Through digitizing and analyzing remote sensing imagery, researchers can identify and analyze geomorphic features of rivers, allowing for effective monitoring and analysis of the process of river morphology evolution.”
References:
Aati, S. and Avouac, J.P. (2020) Optimization of Optical Image Geometric Modeling, Application to Topography Extraction and Topographic Change Measurements Using PlanetScope and SkySat Imagery. Remote Sensing 12(20).
Liu, Y.K., Ma, L.L., Wang, N., Qian, Y.G., Zhao, Y.G., Qiu, S., Gao, C.X., Long, X.X. and Li, C.R. (2020) On-orbit radiometric calibration of the optical sensors on-board SuperView-1 satellite using three independent methods. Optics Express 28(8), 11085-11105.
Point 4:
2.1 Study area (pag. 2): “Covering an area of 11,575 km², the region had an estimated population of 7.34 million in 2019” Are you referring to the River Basin area?
Response 4:
Thanks to the review’s comment. The area and population mentioned here refer to the city of Fuzhou, where the study area is located. The authors regret any misunderstanding caused by their expression error and has made revisions in the text. The modified sentence is as follows:
“Fuzhou City covers an area of 11,575 square kilometers and the population of the area is estimated to be 7.34 million in 2019.”
Point 5:
2.1 Study area (pag. 2): The MQMH section, which is depicted in Fig. 1, is not mentioned in the text.
Response 5:
Thanks to the reviewer for pointing out the issue with the authors' presentation. The authors have added a description of the MQMH section in the manuscript, and the supplemental description is shown as follow:
“MQMH section is the river section between Shuikou Reservoir and HG, the rivers are mostly mountainous type rivers, which are relatively less directly influenced by human activities.”
Point 6:
Fig. 1: The colours of the NG and MW sections are too similar. I would recommend changing the tone. I would also recommend including a small map of China to better understand the location of the river.
Response 6:
Thanks to the reviewer's suggestions, the authors have revised the colors of the NG section and MW section, and added a map of China so that readers can have a clearer idea of the location of the study area.
Point 7:
Table 1: The capital letter in “Sentinel2” is missing.
Response 7:
Thanks to the reviewer for pointing out the errors, and the authors have made corrections in the manuscript.
Point 8:
2.2.2 Model training dataset (pag. 4): “To improve the model’s recognition ability under diverse conditions, we expand the dataset using image enhancement techniques such as random rotation, contrast adjustment, horizontal and vertical flipping, and brightness contrast adjustment” Has anyone already used this method to expand the dataset? (If so, please insert some citations).
Response 8:
Thanks to the suggestion of the reviewer. These pre-processing methods have been already used in the field of computer vision recognition field. For example, Bin, Zhong et al. used similar processing methods in identifying water bodies based on remote sensing images, and the authors have added references in the corresponding positions. The added references are shown as bellow:
Bin, W., Chen, Z.L., Wu, L., Yang, X.H. and Zhou, Y. (2022) SADA-Net: A Shape Feature Optimization and Multiscale Context Information-Based Water Body Extraction Method for High-Resolution Remote Sensing Images. Ieee Journal of Selected Topics in Applied Earth Observations and Remote Sensing 15, 1744-1759.
Zhong, H.F., Sun, Q., Sun, H.M. and Jia, R.S. (2022) NT-Net: A Semantic Segmentation Network for Extracting Lake Water Bodies From Optical Remote Sensing Images Based on Transformer. Ieee Transactions on Geoscience and Remote Sensing 60.
Point 9:
3.1 Remote sensing image pre-processing (pag. 4): “The pre-processing of remote sensing images includes several steps. First, area network leveling is performed using the panchromatic image to match connection points and ensure an error of less than 3 pixels” Please specify that the panchromatic band was only used for Landsat-8 images.
Response 9:
Thanks to the reviewer for pointing out the mistake, the authors have revised it in the manuscript. The revised sentence is shown as follow:
“First, area network leveling is performed using the panchromatic image to match connection points and ensure an error of less than 3 pixels (Landsat-8 images only).”
Point 10:
3.1 Remote sensing image pre-processing (pag. 4): Please explain in the text whether you resampled the sensor bands to the same spatial resolution and if you used the images as level 2 (already atmospherically corrected).
Response 10:
Thanks for the reviewer's comments. In this study, the authors used a deep learning-based image recognition method to identify water bodies. During the water body interpretation process, we utilized multi-band image data from Landsat 5, Landsat 8, and Sentinel-2. To account for the varying resolutions of the bands, we employed the nearest neighbour interpolation method to resample the Landsat image bands to a uniform resolution of 30m and the Sentinel-2 image bands to a uniform resolution of 10m. We have made the necessary revisions in section 3.1. The revised sentence is shown as follow:
“The Landsat and Sentinel-2 sensor bands are then resampled to a resolution of 30m and 10m, respectively, using the nearest neighbour interpolation method. To achieve overall color balance, the hue, saturation, contrast, and brightness are adjusted based on grayscale characteristics of the remote sensing image and reference image.”
Point 11:
3.2 River water interpretation method (pag. 4): “Firstly, neural network is used for automatic recognition of water bodies in the whole area” I do not understand whether the neural networks rely on the spectral signature to perform the interpretation of the image or whether it is a kind of automatic interpretative visualisation.
Response 11:
Thank you for the reviewer's comments. The authors appreciate the reviewer's comments and apologize for the lack of clarity in the previous statement, which may have caused confusion for readers. In this study, the neural network was used to automatically recognize water bodies based on the spectral information of remote sensing images. For Landsat 5 images, the model used information from bands 1, 2, 3, 4, 5, and 7 as inputs. For Landsat 8 images, the model used information from bands 2, 3, 4, 5, 6, and 7 as inputs. For Sentinel-2 images, the model used information from bands 2, 3, 4, 5, 9, and 10 as inputs, with water and non-water labels matched to serve as the outputs, to train the neural network for automatic identification of water bodies. we have made revisions and additions to section 3.2 of the paper to provide a clearer description of the this process. The revised sentences are shown as follow:
“Remote sensing image interpretation is carried out by combining neural network and visual interpretation method in this study. Firstly, the neural network utilizes multi-band data from remote sensing images as inputs, and is trained with water and non-water labels to automatically recognize water bodies. Specifically, the bands 1-5 and 7 of Landsat 5 images, bands 2-7 of Landsat 8 images, and bands 2-5, 9, and 10 of Sentinel-2 images are used as inputs, respectively. And then, based on high-definition remote sensing image, visual interpretation method is used for secondary correction of local areas with poor automatic recognition effect to improve the recognition effect of water bodies. Finally, GIS tools are used to calculate the morphological information of characterized rivers.”
Point 12:
3.3 Beaches and sandbars interpretation method (pag. 6): “Landsat 5 and Landsat 8 images are then extracted using a combination of bands 543, while Sentinel remote sensing images are identified using bands 654” I recommend rewriting the sentence so that it is better understood which bands have been chosen in the RGB display, for each sensor.
Response 12:
Thanks to the reviewer's suggestion, the authors appreciate the reviewer's suggestion and have added corresponding instructions in the text where the expression was missing. The revised sentence is shown as follows:
“Landsat 5 and Landsat 8 images are then extracted using a combination of bands 543(band5 for red, band4 for green, and band3 for blue in RGB), while Sentinel remote sensing images are identified using bands 654(band6 for red, band5 for green, and band4 for blue in RGB).”
Point 13:
3.5.3 Land use transfer matrix (pag. 7): “As a result, the land use transfer matrix is widely used in analyzing land use change” Please insert some references.
Response 13:
Thanks to the reviewer's suggestion, the authors have added the necessary references, including:
Ayalew, A.D., Wagner, P.D., Sahlu, D. and Fohrer, N. (2022) Land use change and climate dynamics in the Rift Valley Lake Basin, Ethiopia. Environmental Monitoring and Assessment 194(10).1-25
Singh, P., Kikon, N. and Verma, P. (2017) Impact of land use change and urbanization on urban heat island in Lucknow city, Central India. A remote sensing based estimate. Sustainable Cities and Society 32, 100-114.
Point 14:
Fig. 5: Missing capital letter in 'water area' in the y-axis.
Response 14:
Thanks to the reviewer to point the mistake, authors have revised it in the manuscript.
Point 15:
Fig. 6: Please explain in the caption that the two circled areas represent the northern and southern part of the NG section.
Response 15:
Thanks to the reviewer’s advice, authors have add the explain in the caption.
Point 16:
Tab. 3: I would suggest moving this table to the supplementary materials.
Response 16:
Thanks to the reviewer’s advice, authors have moved the Table 3 to the supplementary materials.
Point 17:
Fig. 7: I would suggest inserting a map of the entire river to highlight the location of the MQMH section. This also applies to the other sections of the river (Fig. 11, 14, 17). I would also suggest moving the number of cross-sections towards the extremities so that they are more readable (also in the other figures).
Response 17:
Thanks to the reviewer’s advice, authors have revised it in the manuscript. As a result of the article's revision, the numbers of certain figures have been changed. In the revised version, the corresponding figure names are Figures 10, 14, 17, and 21, respectively.
Point 18:
4.3.1 MQMH section (pag. 10): “The distribution of river water, sandbars, and beaches in the MQMH section is depicted in Figure 8 illustrates the distribution of river water, sandbars, and beaches in the MQMH section” Please rewrite this sentence (there are repetitions).
Response 18:
Thanks to the reviewer for pointing out the mistake, the authors have revised the sentence in the manuscript. The revised sentence is showing as follow:
“The distribution of river water, sandbars, and beaches in the MQMH section is depicted in Figure 8.”
Point 19:
Fig. 9: I would suggest using the same three colours you used for Fig. 8 (water=blue; sandbar=green; beach=yellow). This also applies to the other sections of the river (Fig. 13, 16, 19).
Response 19:
Thanks to the reviewer for the suggestions on the optimization of the manuscript's figures. Following the reviewer's recommendations, Figures 9, 13, 16, and 19 have been modified to ensure consistency in color scheme with Figure 8 and to improve readability. As a result of the article's revision, the numbers of certain figures have been changed. In the revised version, the corresponding figure names are Figures 13, 16, 20, and 23, respectively.
Point 20:
Fig. 10: I would suggest moving this figure to the supplementary materials.
Response 20:
Thanks to the suggestion of the reviewer, the authors have moved the Figure 10 to the supplementary materials.
Point 21:
5.3 Impact of river training measures along the river (pag. 19): “The project dredges a total of 4,564,500 m3, and some spur dikes are laid along the improvement section to adjust the flow direction (Figure 20).” Perhaps you meant Fig. 21.
Response 21:
Thanks to the reviewer for pointing out the error, the “Figure 20” should be replaced as “Figure 21”. The authors have revised it in the manuscript.
Point 22:
Fig. 21: I would suggest moving this figure to the supplementary materials.
Response 22:
Thanks to the suggestion of the reviewer, the authors have moved the Figure 21 to the supplementary materials.
Point 23:
Tab. 4: I would suggest moving this table to the supplementary materials.
Response 23:
Thanks to the suggestion of the reviewer, the authors have moved the Table 4 to the supplementary materials.

Reviewer 3 Report
Adding a map of the location in the country and the catchment would help a lot in the interpretation and description. Same about adding some DEM or contour lines to better connect the topography with the different segments of the river.
Although this is a specialized journal and most readers would be familiarized with these indexes, it would be good to provide a description of them and tell what they represent and how different and complementary they are.
The title says “integrated” approach, but the catchment processes are missing as the study is restricted to a limited region adjacent to the river and the river itself only. The catchment perspective is necessary, especially when the discussion is focused on interventions taking place there.
How has the river streamflow changed over time and, specifically, before each satellite image was obtained? It would be convenient to add a graph of this variable for the study period.
Some minor typos were detected.
Author Response
Responses to the reviewer 3
First of all, the authors thank the reviewer very much for the very careful and thorough check of the manuscript. We worked on your corrections, comments and suggestions, and we have addressed most of them in an acceptable way. According to the comments, our replies are listed as following:
Point 1:
Adding a map of the location in the country and the catchment would help a lot in the interpretation and description. Same about adding some DEM or contour lines to better connect the topography with the different segments of the river.
Response 1:
Thanks to the reviewer's suggestion, the authors have added information on the country and region where the study area is located in Figure 1, and supplemented with a DEM map of the study area in Figure 2.

Figure 1 Schematic diagram of the study area

Figure 2 The DEM of the study area
Point 2:
Although this is a specialized journal and most readers would be familiarized with these indexes, it would be good to provide a description of them and tell what they represent and how different and complementary they are.
Response 2:
Thanks to the suggestion of the reviewer, providing a brief description of the indexes used in the study is beneficial for readers to better understand the evaluation of the algorithm's results. The authors have made corresponding modifications to Section 3.4 of the paper based on the reviewer's suggestion. Some of the modified content is presented below:
Accuracy is the proportion of correctly classified samples to the total number of samples, and a value closer to 1 indicates better model performance. It is useful when the classes are well-balanced and the cost of false positives and false negatives is similar. Precision is the proportion of true positive samples to the total number of predicted positive samples, and it is a useful metric when the cost of false positives is high. Recall is the proportion of true positive samples to the total number of actual positive samples, and it measures the ability of the model to identify positive samples. It is useful when the cost of false negatives is high. F1 score is the harmonic mean of precision and recall, and it provides a balanced evaluation of the model's performance. It is particularly useful when the classes are imbalanced. Kappa coefficient is a statistic that measures the agreement between the predicted and actual classifications of a model, taking into account the possibility of agreement by chance. The value of kappa ranges between -1 and 1, with a value of 1 indicating perfect agreement and a value of 0 indicating agreement by chance. Equations (1)-equation (5) can be used to express accuracy, precision, recall, F1 score, and Kappa coefficient, respectively.

Where F1 represents the F1 score; represents the Kappa coefficient; represents the number of positive samples correctly classified; represents the number of negative samples correctly classified; represents the number of positive samples incorrectly classified; and represents the number of negative samples incorrectly classified.
Point 3:
The title says “integrated” approach, but the catchment processes are missing as the study is restricted to a limited region adjacent to the river and the river itself only. The catchment perspective is necessary, especially when the discussion is focused on interventions taking place there.
Response 3:
Thanks for the reviewer's comments. The title of this study is "An integrated approach for analyzing the river morphological evolution based on long-term remote sensing data". The term "integrated" in the title primarily refers to the combination of long-term remote sensing image data and deep learning methods to analyze the morphological evolution in the lower reaches of the Minjiang River. All of the analyses of river morphological evolution in this study are focused on the lower reaches of the Minjiang River. Although the study is limited by data availability and does not provide a comprehensive analysis of the entire river basin, it is still meaningful for exploring the patterns of river morphological evolution in the basin, as well as human activities that may impact the river and potential measures for their mitigation. Therefore, we have revised the title of the paper to more accurately reflect its specific focus, and the revised title is as follows: "An Integrated Approach for Analyzing Morphological Evolution of the Lower Reaches of the Minjiang River Based on Long-term Remote Sensing Data".
Point 4:
How has the river streamflow changed over time and, specifically, before each satellite image was obtained? It would be convenient to add a graph of this variable for the study period.
Response 4:
Thanks to the advice of reviewer. The authors agree with the reviewer that streamflow are crucial factors in the morphological evolution of rivers. In Section 5.1, historical flow and sediment transport data (Figure 21) were presented by the author to discuss the impact of large-scale water conservancy facilities on river morphology. However, including this information in the same subsection as the impact of water conservancy projects may have led readers to overlook this vital information. Thus, the author relocated this information to Section 4.2 and complemented it with flow and sediment transport data from specific years to enhance the data comprehensiveness. Some of the specific modifications made are shown as below:
“Figure 6 presents the annual variations in runoff and sediment transport at the Zhuqi hydrological station from 1986 to 2021. Over this timeframe, the annual runoff displays a fluctuating pattern, with the highest value in 2016 and the lowest in 1991, without a significant overall trend. There are some fluctuations in the annual sediment transport from 1986 to 1992, which decreases significantly after 1993. Moreover, the relationship between flow and sediment during the periods of 1986-1992 and 1993-2021 is shown in Figure 7. A strong linear relationship is observed between annual runoff and sediment transport from 1986 to 1992, with a linear regression R2 value of 0.985. However, after 1993, this relationship decreases, with an R2 value of 0.544. Additionally, the annual sediment transport shows a more obvious decreasing trend under the same annual runoff conditions.”

Figure 6 Annual runoff and sediment transport at Zhuqi station

Figure 7 Correlation between annual runoff and sediment transport at Zhuqi Station

Round 2
Reviewer 3 Report
The manuscript improved and I consider it suitable to be published.